# NRGBoost: Energy-Based Generative Boosted Trees

**João Bravo**
Feedzai
`joao.g.bravo@outlook.com`

## Abstract

Despite the rise to dominance of deep learning in unstructured data domains, tree-based methods such as Random Forests (RF) and Gradient Boosted Decision Trees (GBDT) are still the workhorses for handling discriminative tasks on tabular data. We explore generative extensions of these popular algorithms with a focus on explicitly modeling the data density (up to a normalization constant), thus enabling other applications besides sampling. As our main contribution we propose an energy-based generative boosting algorithm that is analogous to the second-order boosting implemented in popular libraries like XGBoost. We show that, despite producing a generative model capable of handling inference tasks over any input variable, our proposed algorithm can achieve similar discriminative performance to GBDT on a number of real world tabular datasets, outperforming alternative generative approaches. At the same time, we show that it is also competitive with neural-network-based models for sampling. Code is available at `https://github.com/ajoo/nrgboost`.

## 1 Introduction

Generative models have achieved tremendous success in computer vision and natural language processing, where the ability to generate synthetic data guided by user prompts opens up many exciting possibilities. While generating synthetic table records does not necessarily enjoy the same wide appeal, this problem has still received considerable attention as a potential avenue for bypassing privacy concerns when sharing data. Estimating the data density, $p(\mathbf{x})$, is another typical application of generative models which enables a host of different use cases that can be particularly interesting for tabular data. Unlike discriminative models, which are trained to perform inference over a single target variable, density models can be used more flexibly for inference over different variables or for out-of-distribution detection. They can also handle inference with missing data in a principled way by marginalizing over unobserved variables.

The development of generative models for tabular data has mirrored its progression in computer vision, with many of its deep learning approaches being adapted to the tabular domain (Jordon et al., 2018; Xu et al., 2019; Engelmann & Lessmann, 2021; Fan et al., 2020; Zhao et al., 2021; Kotelnikov et al., 2023). Unfortunately, these methods are only useful for sampling as they either do not model the density explicitly or can not evaluate it efficiently due to intractable marginalization over high-dimensional latent variable spaces. Furthermore, despite growing in popularity, deep learning has still failed to displace tree-based ensemble methods as the tool of choice for handling tabular discriminative tasks, with gradient boosting still being found to outperform neural-network-based methods in many real world datasets (Grinsztajn et al., 2022; Borisov et al., 2022a).

While there have been recent efforts to extend the success of tree-based models to generative modeling (Correia et al., 2020; Wen & Hang, 2022; Nock & Guillame-Bert, 2022; Watson et al., 2023; Nock & Guillame-Bert, 2023; Jolicoeur-Martineau et al., 2024; McCarter, 2024), direct extensions of Random Forests (RF) and Gradient Boosted Decision Trees (GBDT) are still missing. We try to address this gap, seeking to keep the general algorithmic structure of these popular algorithms but replacing the optimization of their discriminative objective with a generative counterpart. Our main contributions in this regard are:

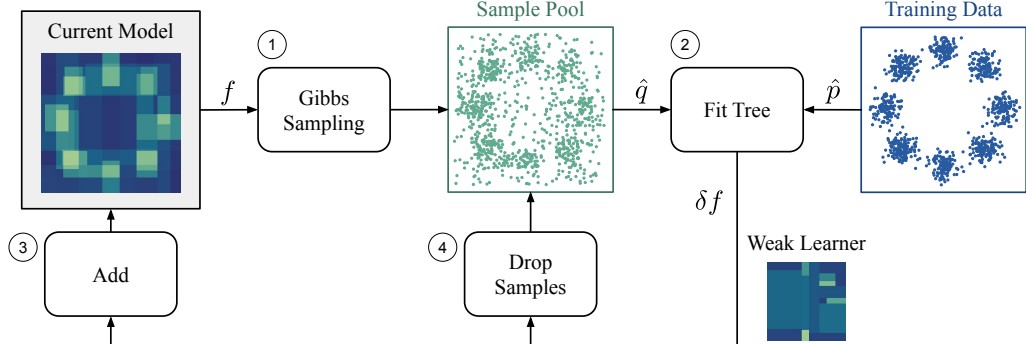

Figure 1: Overview of an NRGBoost training iteration: 1) Draw new samples from an ensemble of trees representing a current energy function, $f$, and add them to the sample pool. 2) Fit a new weak learner (tree) representing a model update, $\delta f$. 3) Update the model. 4) Use rejection sampling to discard samples from the sample pool that conform poorly to the new model.

- We propose NRGBoost, a novel energy-based generative boosting model that is trained to maximize a local second-order approximation to the log-likelihood at each boosting round.

- We propose an amortized sampling approach that significantly reduces the training time of NRGBoost, which, like other energy-based models, is often dominated by the Markov chain Monte Carlo sampling required for training.

- We explore the use of bagged ensembles of Density Estimation Trees (Ram & Gray, 2011) with feature subsampling as a generative counterpart to Random Forests.

The longstanding popularity of GBDT models in machine learning practice can, in part, be attributed to the strength of its empirical results and the efficiency of its existing implementations. We therefore focus on an experimental evaluation in real world datasets spanning a range of use cases, number of samples and features. On smaller datasets, our implementation of NRGBoost can be trained in a few minutes on a typical consumer CPU and achieves similar discriminative performance to a standard GBDT model, while remaining competitive with other generative models for sampling.

## 2 ENERGY BASED MODELS

An Energy-Based Model (EBM) parametrizes the logarithm of a probability density function directly (up to an unspecified normalizing constant):

$$q_f(\mathbf{x}) = \frac{\exp\left(f(\mathbf{x})\right)}{Z[f]}. \tag{1}$$

Here, $f(\mathbf{x}) : \mathcal{X} \rightarrow \mathbb{R}$ is a real function over the input domain.[1] We will avoid introducing any parametrization and instead treat the function $f \in \mathcal{F}(\mathcal{X})$, lying in an appropriate function space over the input domain, as our model parameter directly. $Z[f] = \sum_{\mathbf{x} \in \mathcal{X}} \exp\left(f(\mathbf{x})\right)$, known as the partition function, is then a functional of $f$ giving us the necessary normalizing constant.

This is the most flexible way one could represent a probability density function, making essentially no compromises on its structure. It can seamlessly handle input domains comprising a mixture of continuous and categorical features, making it a natural choice for tabular data. The downside to this is that, for most interesting choices of $\mathcal{F}$, computing the normalizing constant is intractable which makes training these models difficult. Their unnormalized nature, however, does not prevent EBMs from being useful in a number of applications besides sampling, such as performing joint inference over small enough groups of variables. For example, by splitting the input domain into a

---

[1]We will assume that $\mathcal{X}$ is finite and discrete to simplify the notation and exposition but everything is applicable to bounded continuous input spaces, replacing the sums with integrals as appropriate.

set of observed variables, $\mathbf{x}_o$, and unobserved variables, $\mathbf{x}_u$, we have

$$q_f(\mathbf{x}_u|\mathbf{x}_o) = \frac{\exp\left(f(\mathbf{x}_u, \mathbf{x}_o)\right)}{\sum_{\mathbf{x}_u'} \exp\left(f(\mathbf{x}_u', \mathbf{x}_o)\right)}, \tag{2}$$

which only involves normalizing (i.e., computing a softmax) over all of the possible values of the unobserved variables. EBMs can also handle inference with missing data in a principled manner by marginalizing over unobserved variables. Furthermore, for anomaly or out-of-distribution detection, knowledge of the normalizing constant is not necessary.

One common way to train an energy-based model to approximate a data generating distribution, $p(\mathbf{x})$, is to minimize the Kullback-Leibler divergence between $p$ and $q_f$, or equivalently, maximize the expected log-likelihood functional:

$$L[f] = \mathbb{E}_{\mathbf{x}\sim p}\log q_f(\mathbf{x}) = \mathbb{E}_{\mathbf{x}\sim p}f(\mathbf{x}) - \log Z[f]. \tag{3}$$

This optimization is typically carried out by gradient descent over the parameters of $f$. However, due to the intractability of the partition function, one must rely on Markov chain Monte Carlo (MCMC) sampling to estimate the gradients (Song & Kingma, 2021).

## 3 NRGBOOST

Expanding the increase in log-likelihood in Equation 3 due to a variation $\delta f$ around an energy function $f$ up to second order (see Appendix A), we have

$$L[f + \delta f] - L[f] \approx \mathbb{E}_{\mathbf{x}\sim p}\delta f(\mathbf{x}) - \mathbb{E}_{\mathbf{x}\sim q_f}\delta f(\mathbf{x}) - \frac{1}{2}\mathrm{Var}_{\mathbf{x}\sim q_f}\delta f(\mathbf{x}) =: \Delta L_f[\delta f]. \tag{4}$$

The $\delta f$ that maximizes this quadratic approximation should thus have a large positive difference between the expected value under the data and under $q_f$, while having low variance under $q_f$. Note that, just like the original log-likelihood, this Taylor expansion is invariant to adding a constant to $\delta f$ (i.e., $\Delta L_f[\delta f + c] = \Delta L_f[\delta f]$ for a constant function $c$). This means that, in maximizing Equation 4, we can restrict our attention to functions that have zero expectation under $q_f$, in which case we can simplify $\Delta L_f$ as

$$\Delta L_f[\delta f] = \mathbb{E}_{\mathbf{x}\sim p}\delta f(\mathbf{x}) - \frac{1}{2}\mathbb{E}_{\mathbf{x}\sim q_f}\delta f^2(\mathbf{x}). \tag{5}$$

We can thus formulate a boosting algorithm that, at each boosting iteration, $t$, improves upon a current energy function, $f_t$, by finding an optimal step, $\delta f_t^*$, that maximizes $\Delta L_{f_t}$:

$$\delta f_t^* = \arg\max_{\delta f\in\mathcal{H}_t} \Delta L_{f_t}[\delta f], \tag{6}$$

where $\mathcal{H}_t$ is an appropriate space of functions (satisfying $\mathbb{E}_{\mathbf{x}\sim q_{f_t}}\delta f(\mathbf{x}) = 0$ if Equation 5 is used). The solution to this problem can be interpreted as a Newton step in the space of energy functions. Because, for an EBM, the Fisher information matrix for the energy function and the Hessian of the expected log-likelihood are the same, we can also interpret this solution as a natural gradient step (see Appendix A). This approach is analogous to the second-order step implemented in modern discriminative gradient boosting libraries such as XGBoost (Chen & Guestrin, 2016) and LightGBM (Ke et al., 2017) and which can be traced back to Friedman et al. (2000).

In updating the current iterate, $f_{t+1} = f_t + \alpha_t \cdot \delta f_t^*$, we scale $\delta f_t^*$ by an additional scalar step-size $\alpha_t$. This can be interpreted as a globalization strategy to account for the fact that the quadratic approximation in Equation 4 is not necessarily valid over large steps in function space. A common strategy in nonlinear optimization would be to select $\alpha_t$ via a line search based on the original log-likelihood. Common practice in discriminative boosting, however, is to interpret this step-size as a regularization parameter and to select a fixed value in $]0, 1]$, with (more) smaller steps typically outperforming fewer larger ones when it comes to generalization. We choose to adopt a hybrid strategy, first selecting an optimal step-size by line search and then shrinking it by a fixed factor. We find that this typically accelerates convergence, allowing the algorithm to take comparatively larger steps that increase the likelihood in the initial phase of boosting.

For a starting point, $f_0$, we can choose the logarithm of any probability distribution over $\mathcal{X}$, as long as it is easy to evaluate. Sensible choices are a uniform distribution (i.e., $f \equiv 0$), the product of marginals for the training set or any mixture distribution between these two. In Figure 2 we show an example of NRGBoost starting from a uniform distribution and learning a toy 2D data density.

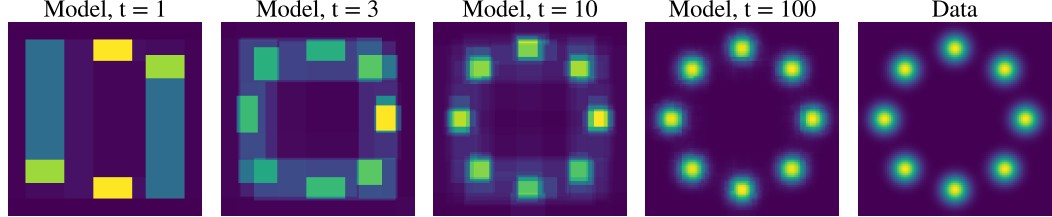

Figure 2: Density learned by NRGBoost at different boosting iterations (1, 3, 10 and 100), starting from a uniform distribution. The data distribution is depicted on the right (see Appendix D.1 for details). Weak learners are piecewise constant functions given by binary trees with 16 leaves.

## 3.1 WEAK LEARNERS

As a weak learner we will consider piecewise constant functions defined by binary trees over the input space. Letting $\bigcup_{j=1}^{J} X_j = \mathcal{X}$ be the partitioning of the input space induced by the leaves of a binary tree whose internal nodes represent a split along one dimension into two disjoint partitions, we take $\mathcal{H}$ as the set of functions such as

$$\delta f(\mathbf{x}) = \sum_{j=1}^{J} w_j \mathbf{1}_{X_j}(\mathbf{x}) \,. \tag{7}$$

Here, $\mathbf{1}_X$ denotes the indicator function of a subset $X$ and $w_j$ are values associated with each leaf $j \in [1..J]$. In a standard decision tree, these values would typically encode an estimate of $p(y|\mathbf{x} \in X_j)$, with $y$ being a special *target* variable that is never considered for splitting. In our generative approach, they encode unconditional densities (or more precisely energies) over each leaf's support and every variable can be used for splitting. Note that the $\delta f$ functions are thus parametrized by the values, $w_j$, as well as the structure of the tree and the variables and thresholds for the split at each node, which ultimately determine the $X_j$. We omit these dependencies for brevity.

By substituting the definition from Equation 7 into our objective (Equation 5) we get the following optimization problem to find the optimal binary tree:

$$\max_{w_1,\ldots,w_J,X_1,\ldots,X_J} \quad \sum_{j=1}^{J} \left( w_j P(X_j) - \frac{1}{2} w_j^2 Q_f(X_j) \right)$$
$$\text{s.t.} \quad \sum_{j=1}^{J} w_j Q_f(X_j) = 0 \,, \tag{8}$$

where $P(X_j)$ and $Q_f(X_j)$ denote the probability of the event $\mathbf{x} \in X_j$ under the respective distribution and the constraint ensures that $\delta f$ has zero expectation under $q_f$. With respect to the leaf weights, this is a quadratic program whose optimal solution and objective values are respectively given by

$$w_j^* = \frac{P(X_j)}{Q_f(X_j)} - 1 \,, \qquad \Delta L_f^*(X_1,\ldots,X_J) = \frac{1}{2} \left( \sum_{j=1}^{J} \frac{P^2(X_j)}{Q_f(X_j)} - 1 \right) \,. \tag{9}$$

Because carrying out the maximization of this optimal value over the tree structure that determines the $X_j$ is hard, we approximate its solution by greedily growing a tree that maximizes it when considering how to split each node individually. A parent leaf with support $X_P$ is thus split into two child leaves, with disjoint support $X_L \cup X_R = X_P$, so as to maximize over all possible partitionings along a single dimension, $\mathcal{P}(X_P)$, the following objective:

$$\max_{X_L,X_R \in \mathcal{P}(X_P)} \frac{P^2(X_L)}{Q_f(X_L)} + \frac{P^2(X_R)}{Q_f(X_R)} - \frac{P^2(X_P)}{Q_f(X_P)} \,. \tag{10}$$

Note that, when using parametric weak learners, computing a second-order step would typically involve solving a linear system with a full Hessian. As we can see, this is not the case when the

weak learners are binary trees, where the optimal value to assign to a leaf $j$ does not depend on any information from other leaves and, likewise, the optimal objective value is a sum of terms, each depending only on information from a single leaf. This would not have been the case had we tried to optimize the log-likelihood functional in Equation 3 directly, instead of its quadratic approximation.

## 3.2 AMORTIZED SAMPLING

To compute the leaf values in Equation 9 and the splitting criterion in Equation 10 we would have to know $P(X)$ and be able to compute $Q_f(X)$, which is infeasible due to the intractable normalization constant. We therefore estimate these quantities, with recourse to empirical data for $P(X)$ and to samples approximately drawn from the model with MCMC. Because, even if the input space is not partially discrete, $f$ is still discontinuous and constant almost everywhere, we can not use gradient based samplers. We therefore rely on Gibbs sampling, which only requires evaluating each $f_t$ along one dimension at a time, while keeping all others fixed. Importantly, this can be computed efficiently for a tree by traversing it only once. However, since our energy function at boosting iteration $t$ is given by a sum of $t$ trees, this computation scales linearly with the iteration number. As a result, the total computational cost of sampling, over the course of training a model, scales quadratically in the number of boosting iterations, thus precluding us from training models with a large number of trees.

In order to reduce the burden associated with this sampling, which dominates the runtime of training the model, we propose a new sampling approach that leverages the cumulative nature of boosting. The intuition behind this approach is that the set of samples used in the previous boosting round are (approximately) drawn from a distribution that is already close to the current model distribution. It could, therefore, be helpful to keep some of those samples, especially those that conform best to the current model. Rejection sampling allows us to do just that. The boosting update in terms of the densities takes the following multiplicative form:

$$q_{t+1}(\mathbf{x}) = k_t \, q_t(\mathbf{x}) \exp\left(\alpha_t \delta f_t(\mathbf{x})\right) . \tag{11}$$

Here, $k_t$ is an unknown multiplicative constant and since $\delta f_t$ is given by a tree, we can easily bound the exponential factor by finding the leaf with the largest value. To sample from $q_{t+1}$, we can therefore use the previous model, $q_t$, as a proposal distribution for which we already have a set of samples. We retain each of these previous samples, $\mathbf{x}$, with an acceptance probability of

$$p_{accept}(\mathbf{x}) = \exp\left[\alpha_t \left(\delta f_t(\mathbf{x}) - \max_{\mathbf{x}} \delta f_t(\mathbf{x})\right)\right] , \tag{12}$$

noting that knowledge of the normalizing constant, $k_t$, is not necessary for its computation.

Our proposed sampling strategy is to maintain a fixed-size pool of approximate samples from the model. At the end of each boosting round, we use rejection sampling to remove samples from the pool and draw new samples from the updated model using Gibbs sampling. Figure 1 depicts a representation of the training loop with this amortized sampling approach. Note that $q_0$ is typically a simple model for which we can both directly evaluate the desired quantities (i.e., $Q_0(X)$ for a given partition $X$) and efficiently draw exact samples. As such, no samples are required for the first round of boosting. For the second round, we can initialize the sample pool by drawing exact samples from $q_1$ with rejection sampling, using $q_0$ as a proposal distribution.

This amortized sampling approach works better when either the range of $\delta f_t$ or the step sizes $\alpha_t$ are small, as this leads to larger acceptance probabilities. Note that, in practice, it can be helpful to independently refresh a fixed fraction of samples, $p_{refresh}$, at each round of boosting in order to encourage a more diverse set of samples between rounds. This can be accomplished by keeping each sample with a probability $p_{accept}(\mathbf{x})(1 - p_{refresh})$ instead. Please refer to Algorithm 1 in Appendix A.3 for a high level algorithmic description of the training process with amortized sampling.

## 3.3 REGULARIZATION

The simplest way to regularize a boosting model is to stop training when overfitting is detected by monitoring a suitable performance metric on a validation set. For NRGBoost this could be the increase in log-likelihood at each boosting round (see Appendix A.1). However, estimating this quantity would require drawing additional validation samples from the model. A viable alternative validation strategy which needs no additional samples is to simply monitor a discriminative performance metric. This amounts to monitoring the quality of $q_f(x_i|\mathbf{x}_{-i})$ instead of the full $q_f(\mathbf{x})$.

Besides early stopping, the binary trees themselves can be regularized by limiting the depth or total number of leaves in each tree. Additionally, we can rely on other strategies, such as disregarding splits that would result in a leaf with too little training data, $P(X)$, model data, $Q_f(X)$, volume, $V(X)$, or too high of a ratio between training and model data, $P(X)/Q_f(X)$. We found the latter to be the most effective of these, not only yielding better generalization performance than other approaches, but also having the added benefit of allowing us to lower bound the acceptance probability of our rejection sampling scheme. Furthermore, as we show in Appendix A.2, limiting this ratio guarantees that a small enough step size produces an increase in training likelihood at each boosting round.

## 4    RELATED WORK

**Generative Boosting**    Most prior work on generative boosting focuses on unstructured data and the use of parametric weak learners and is split between two approaches: (i) Additive methods that model the density function as an additive mixture of weak learners such as Rosset & Segal (2002); Tolstikhin et al. (2017). (ii) Those that take a multiplicative approach, modeling the density function as an unnormalized product of weak learners. The latter is equivalent to the energy-based approach that writes the energy function (log-density) as an additive sum of weak learners. Welling et al. (2002), in particular, also approaches boosting from the point of view of functional optimization of the log-likelihood or the logistic loss of an energy-based model. However, it relies on a first order local approximation of the objective given that it focuses on parametric weak learners, such as restricted Boltzmann machines, for which a second-order step would be impractical.

Another more direct multiplicative boosting framework was first proposed by Tu (2007). At each boosting round, a discriminative classifier is trained to distinguish between empirical data and data generated by the current model, by estimating the likelihood ratio $p(\mathbf{x})/q_t(\mathbf{x})$. This estimated ratio is used as a direct multiplicative factor to update a current model, $q_t$. In ideal conditions, this greedy procedure would converge in a single iteration (when using a step size of 1). While Tu (2007) does not prescribe a particular choice of classifier, Grover & Ermon (2018) proposes a similar concept where the ratio is estimated based on a variational bound for an $f$-divergence and Cranko & Nock (2019) provides additional analysis on this method. We note that the main difference between this greedy approach and NRGBoost is that the latter attempts to update the current density proportionally to an exponential of the likelihood ratio, $\exp\left(\alpha_t \cdot p(x)/q_t(x)\right)$, instead of $\left(p(\mathbf{x})/q_t(\mathbf{x})\right)^{\alpha_t}$ directly. In Appendix C we explore the differences between NRGBoost and this approach, when it is adapted to use trees as weak learners.

**Tabular Energy-Based Models**    Ma et al. (2024) proposes reinterpreting the logits of a TabPFN classifier (Hollmann et al., 2022) as an energy function over the input space. Similarly, Margeloiu et al. (2024) relies on auxiliary binary classification tasks, reinterpreting the logits of the classifiers as class-conditional energy functions. These approaches, while not as principled as maximum likelihood, avoid the need for MCMC sampling and were shown to work well empirically. However, their reliance on TabPFN can limit their applicability to larger datasets. Furthermore, both approaches rely on having a categorical *target* variable, which further limits their applicability and could introduce an inherent bias given that this variable is not treated in the same way as the others.

**Tree-Based Density Models**    Density Estimation Trees (DET) were proposed by Ram & Gray (2011) as an alternative to histograms and kernel density estimation. They model the density function as a constant value over the support of each leaf in a binary tree, $q(\mathbf{x}) = \sum_{j=1}^{J} \frac{\hat{P}(X_j)}{V(X_j)} \mathbf{1}_{X_j}(\mathbf{x})$, with $V(X)$ denoting the volume of $X$. The partitioning of the input space is determined directly by greedy optimization of a generative objective (ISE). Despite allowing for efficient exact sampling, DETs have received little attention as generative models. In Appendix B, we explore extensions of this method to maximum likelihood estimation. In our experiments, we also test the natural idea to ensemble DET models through bagging, leading to a generative algorithm analogous to Random Forests.

Other authors have proposed similar tree-based density models (Nock & Guillame-Bert, 2022) or additive mixtures of tree-based models (Correia et al., 2020; Wen & Hang, 2022; Watson et al., 2023). Distinguishing features of some of these alternative approaches are: (i) Not relying on a density estimation goal to drive the partitioning of the input space. Correia et al. (2020) leverages a standard discriminative Random Forest, therefore giving special treatment to a particular input variable whose

conditional estimation drives the choice of partitions and Wen & Hang (2022) proposes using a mid-point random tree partitioning. (ii) Leveraging more complex models for the data in the leaf of a tree instead of a uniform density (Correia et al., 2020; Watson et al., 2023). This approach can allow for the use of trees that are more representative with a smaller number of leaves. (iii) Nock & Guillame-Bert (2022) and Watson et al. (2023) both propose generative adversarial frameworks where the generator and discriminator are both a tree or an ensemble of trees respectively. While this also leads to an iterative approach, unlike with boosting or bagging the new model trained at each round doesn't add to the previous one but replaces it instead.

Nock & Guillame-Bert (2023) proposes a different ensemble approach where trees do not have their own leaf values that are added or multiplied to produce a final density, but instead collectively define the partitioning of the input space. To train such models the authors propose a framework where, rather than adding a new tree to the ensemble at every iteration, the model is initialized with a fixed number of tree root nodes and each iteration adds a split to an existing leaf node. Jolicoeur-Martineau et al. (2024) proposes diffusion and conditional flow matching approaches where a tree-based model (e.g., GBDT) is used to learn the score function or vector field. However, these approaches do not support efficient density estimation. McCarter (2024) leverages the autoregressive framework of TabMT (Gulati & Roysdon, 2024), replacing transformers with XGBoost based predictors. While this approach is capable of density estimation, it requires training one predictor (consisting of one or more XGBoost models) per input variable. Furthermore, these models are trained on datasets created from multiple copies of the original dataset with different missing data patterns, which could limit their applicability to larger datasets.

## 5 EXPERIMENTS

We evaluate NRGBoost on five tabular datasets from the UCI Machine Learning Repository (Dheeru & Karra Taniskidou, 2017): Abalone (AB), Physicochemical Properties of Protein Tertiary Structure (PR), Adult (AD), MiniBooNE (MBNE) and Covertype (CT) as well as the California Housing (CH) dataset available through scikit-learn (Pedregosa et al., 2011). We also include a downsampled version of MNIST (by 2x along each dimension), which allows us to visually assess the quality of individual samples, something that is generally difficult with structured tabular data. More details about these datasets are given in Appendix D.1.

We split our experiments into two sections, the first to evaluate the quality of density models directly on a single-variable inference task and the second to compare the performance of our proposed model on sampling use cases against more specialized models.

### 5.1 SINGLE-VARIABLE INFERENCE

We test the ability of a generative model, trained to learn the density over all input variables, $q(\mathbf{x})$, to infer the value of a single one given the others. Specifically, we evaluate the quality of its estimate of $q(x_i|\mathbf{x}_{-i})$. For this purpose we pick $x_i = y$ as the original target of the dataset, noting that the models that we train do not treat this variable in any special way, except when selecting the best model in validation. As such, we would expect that a model's performance in inference over this particular variable is indicative of its strength on any other single-variable inference task and also indicative of the quality of the full $q(\mathbf{x})$, from which the conditional probability estimate is derived.

We use XGBoost (Chen & Guestrin, 2016) as a baseline for what should be achievable by a strong discriminative model. Note that this model is trained to maximize the discriminative likelihood, $\mathbb{E}_{\mathbf{x} \sim p} \log q(x_i|\mathbf{x}_{-i})$, not wasting model capacity in learning other aspects of the full data distribution. Because generative models provide an estimate of the full conditional distribution for the regression datasets, rather than a single point estimate like XGBoost, we also include NGBoost (Duan et al., 2020) as another discriminative baseline that can estimate the full $q(x_i|\mathbf{x}_{-i})$ on these datasets. Note, however, that NGBoost relies on a parametric assumption about this conditional distribution, that needs to hold for any $\mathbf{x}_{-i}$. We use the default assumption of a Normal distribution for all datasets.

We compare NRGBoost against a bagging ensemble of DET models (Ram & Gray, 2011), which we call *Density Estimation Forests* (DEF), trained for minimizing either ISE or KL divergence (see Appendix B). We also include two other tree-based generative baselines: RFDE (Wen & Hang, 2022)

Table 1: Discriminative performance of different methods at inferring the value of a target variable. We use $R^2$ for regression tasks, AUC for binary classification and accuracy for multiclass classification. The reported values are means and standard errors over 5 cross-validation folds. The best *generative method* for each dataset is highlighted in **bold** and other methods that are not significantly worse (as determined by a paired $t$-test at a 95% confidence level) are underlined.

| | $R^2$ ↑ | | | AUC ↑ | | Accuracy ↑ | |
| --- | --- | --- | --- | --- | --- | --- | --- |
| | AB | CH | PR | AD | MBNE | MNIST | CT |
| XGBoost | 0.552 ±0.035 | 0.849 ±0.009 | 0.678 ±0.004 | 0.927 ±0.000 | 0.987 ±0.000 | 0.976 ±0.002 | 0.971 ±0.001 |
| NGBoost | 0.546 ±0.040 | 0.829 ±0.009 | 0.621 ±0.005 | - | - | - | - |
| RFDE | 0.071 ±0.096 | 0.340 ±0.004 | 0.059 ±0.007 | 0.862 ±0.002 | 0.668 ±0.008 | 0.302 ±0.010 | 0.679 ±0.002 |
| ARF | 0.531 ±0.032 | 0.758 ±0.009 | 0.591 ±0.007 | 0.893 ±0.002 | 0.968 ±0.001 | -* | 0.938 ±0.005 |
| DEF (ISE) | 0.467 ±0.037 | 0.737 ±0.008 | 0.566 ±0.002 | 0.854 ±0.003 | 0.653 ±0.011 | 0.206 ±0.011 | 0.790 ±0.003 |
| DEF (KL) | 0.482 ±0.027 | 0.801 ±0.008 | 0.639 ±0.004 | 0.892 ±0.001 | 0.939 ±0.001 | 0.487 ±0.007 | 0.852 ±0.002 |
| NRGBoost | **0.547 ±0.036** | **0.850 ±0.011** | **0.676 ±0.009** | **0.920 ±0.001** | **0.974 ±0.001** | **0.966 ±0.001** | **0.948 ±0.001** |

> * Due to the discrete nature of the MNIST dataset and the fact that the ARF algorithm tries to fit continuous distributions for numerical variables at the leaves, we could not obtain a reasonable density model for this dataset.

and ARF (Watson et al., 2023). The former allows us to gauge the impact of the guided partitioning used in DEF models over a random partitioning of the input space.

We use random search to tune the hyperparameters of XGBoost and NGBoost and a grid search to tune the most important hyperparameters of each generative density model. We employ 5-fold cross-validation, repeating the hyperparameter tuning on each fold. For the full details of the experimental protocol please refer to Appendix D.

We find that NRGBoost outperforms the remaining generative models (see Table 1), even achieving comparable performance to XGBoost on the smaller datasets and with a small gap on the three larger ones (MBNE, MNIST and CT). Furthermore, in Appendix E.3 we show that, for the same tasks but in the presence of a missing covariate, NRGBoost can outperform XGBoost relying on popular imputation strategies due to its principled handling of missing data through marginalization.

## 5.2 SAMPLING

We evaluate two different aspects of the quality of generated samples: their utility for training a machine learning model and how distinguishable they are from real data. In addition to the previous ARF and DEF (KL) baselines, we compare against TVAE (Xu et al., 2019) and TabDDPM (Kotelnikov et al., 2023), two deep-learning-based generative models, as well as Forest-Flow (Jolicoeur-Martineau et al., 2024), a tree-based conditional flow matching model.

**Machine Learning Efficiency** Machine learning (ML) efficiency has been a popular way to measure the quality of generative models for sampling (Xu et al., 2019; Kotelnikov et al., 2023; Borisov et al., 2022b). It relies on using samples from the model to train a discriminative model which is then evaluated on real data, and is thus similar to the single-variable inference performance that we use to compare density models in Section 5.1. In fact, if the discriminative model is flexible enough, one would expect it to recover the generator's $q(y|\mathbf{x})$, and therefore its performance, in the limit where infinite generated data is used to train it.

We use XGBoost as the discriminative model and train it on the same number of synthetic samples as in the original training data. For the density models, we generate the samples from the best model found in the previous section. For TVAE and TabDDPM, we select their hyperparameters by evaluating the ML efficiency in the real validation set (full details of the hyperparameter tuning are provided in Appendix D.3). Note that this leaves these models at a potential advantage since the hyperparameter selection is based on the metric that is being evaluated.

We repeat all experiments 5 times, with different generated datatsets from each model and report the performance of the discriminative model in Table 2. We find that NRGBoost and TabDDPM alternate as the best-performing model depending on the dataset (with two inconclusive cases), and NRGBoost

Table 2: Performance of an XGBoost model trained on synthetic data and on real data (for reference). For consistency, we use the same discriminative metrics as in Table 1 for evaluating its performance. Reported values are the averages and standard errors over 5 synthetic datasets generated by the same generative model. The best generative method for each dataset is highlighted in **bold** and methods that are not significantly worse (as determined by a $t$-test at a 95% confidence level) are underlined.

| | $R^2 \uparrow$ | | | AUC $\uparrow$ | | Accuracy $\uparrow$ | |
|---|---|---|---|---|---|---|---|
| | AB | CH | PR | AD | MBNE | MNIST | CT |
| Real Data | 0.554 | 0.838 | 0.682 | 0.927 | 0.987 | 0.976 | 0.972 |
| TVAE | 0.483 ±0.006 | 0.758 ±0.005 | 0.365 ±0.005 | 0.898 ±0.001 | 0.975 ±0.000 | 0.770 ±0.009 | 0.750 ±0.002 |
| TabDDPM | **0.539** ±0.018 | **0.807** ±0.005 | **0.596** ±0.007 | 0.910 ±0.001 | **0.984** ±0.000 | - | 0.818 ±0.001 |
| Forest-Flow | 0.418 ±0.019 | 0.716 ±0.003 | 0.412 ±0.009 | 0.879 ±0.003 | 0.964 ±0.001 | 0.224 ±0.010 | 0.705 ±0.002 |
| ARF | 0.504 ±0.020 | 0.739 ±0.003 | 0.524 ±0.003 | 0.901 ±0.001 | 0.971 ±0.001 | 0.908 ±0.002 | 0.848 ±0.002 |
| DEF (KL) | 0.450 ±0.013 | 0.762 ±0.006 | 0.498 ±0.007 | 0.892 ±0.001 | 0.943 ±0.002 | 0.230 ±0.028 | 0.753 ±0.002 |
| NRGBoost | 0.528 ±0.016 | 0.801 ±0.001 | 0.573 ±0.008 | **0.914** ±0.001 | 0.977 ±0.001 | **0.959** ±0.001 | **0.895** ±0.001 |

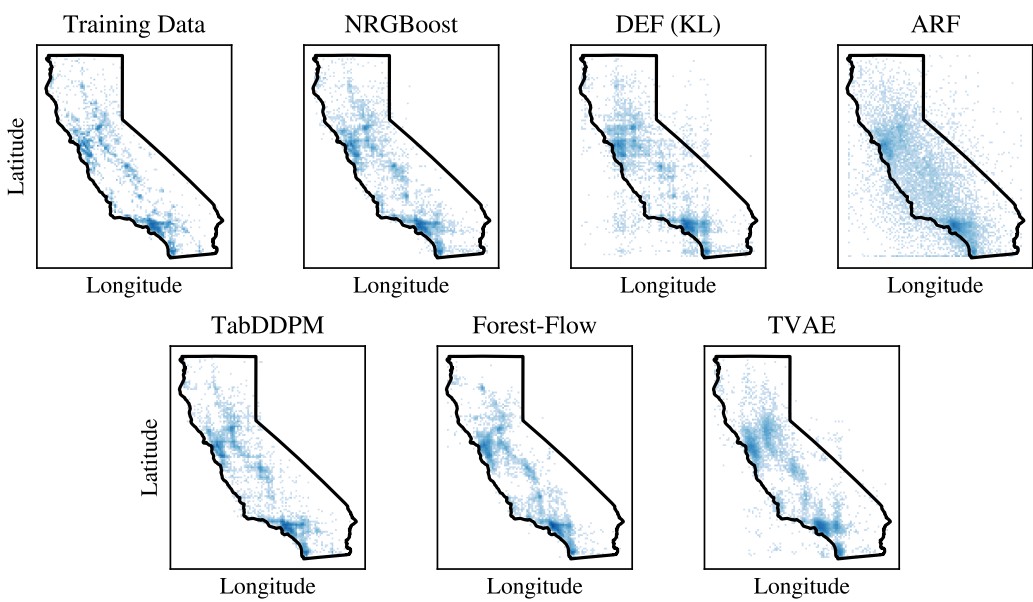

Figure 3: Joint histogram for the latitude and longitude for the California Housing dataset.

is never ranked lower than second on any dataset. We also note that, despite our best efforts with additional manual tuning, we could not obtain a reasonable model with TabDDPM on MNIST.

**Discriminator Measure**    Similar to Borisov et al. (2022b) we test the capacity of a discriminative model to distinguish between real and generated data. We use the original validation set as the real part of the training data for the discriminator in order to avoid benefiting generative methods that overfit their original training set. A new validation set is carved out of the original test set (20%) and used to tune the hyperparameters of an XGBoost model which we use as our discriminator. We report the AUC of this model on the remainder of the test data in Table 3, which shows that NRGBoost outperforms other methods except on the PR (where the result is inconclusive) and MBNE datasets.

In Appendix E.1 we report a Wasserstein distance estimate computed using a similar setup to Jolicoeur-Martineau et al. (2024) as an alternative measure of the statistical distance between real and model distributions. This evaluation also shows NRGBoost as ranking the best on average across datasets. Qualitatively speaking, its samples also look visually similar to the real data in both the California and MNIST datasets (see Figures 3 and 4).

| Training Data | NRGBoost | Forest-Flow | ARF | TVAE |
|---|---|---|---|---|



Figure 4: Downsampled MNIST samples generated by the best generative models on this dataset. Despite being a simple dataset that would pose no challenges to image models, it is hard for tabular generative models due to the high dimensionality and complex structure of correlations between features. We find NRGBoost to be the only tabular model that is able to generate passable samples.

Table 3: AUC of an XGBoost model trained to distinguish real from generated data (lower is better). Reported values are the averages and standard errors over 5 synthetic datasets generated by the same model. The best generative method for each dataset is highlighted in **bold** and methods that are not significantly worse (as determined by a $t$-test at a 95% confidence level) are underlined.

|  | AB | CH | PR | AD | MBNE | MNIST | CT |
|---|---|---|---|---|---|---|---|
| TVAE | $0.971_{\pm0.004}$ | $0.834_{\pm0.006}$ | $0.940_{\pm0.002}$ | $0.898_{\pm0.001}$ | $1.000_{\pm0.000}$ | $1.000_{\pm0.000}$ | $0.999_{\pm0.000}$ |
| TabDDPM | $0.818_{\pm0.015}$ | $0.667_{\pm0.005}$ | $\mathbf{0.628_{\pm0.004}}$ | $0.604_{\pm0.002}$ | $\mathbf{0.789_{\pm0.002}}$ | - | $0.915_{\pm0.007}$ |
| Forest-Flow | $0.987_{\pm0.002}$ | $0.926_{\pm0.002}$ | $0.885_{\pm0.002}$ | $0.932_{\pm0.002}$ | $1.000_{\pm0.000}$ | $1.000_{\pm0.000}$ | $0.985_{\pm0.001}$ |
| ARF | $0.975_{\pm0.005}$ | $0.973_{\pm0.004}$ | $0.795_{\pm0.008}$ | $0.992_{\pm0.000}$ | $0.998_{\pm0.000}$ | $1.000_{\pm0.000}$ | $0.989_{\pm0.001}$ |
| DEF (KL) | $0.823_{\pm0.013}$ | $0.751_{\pm0.008}$ | $0.877_{\pm0.002}$ | $0.956_{\pm0.002}$ | $1.000_{\pm0.000}$ | $1.000_{\pm0.000}$ | $0.999_{\pm0.000}$ |
| NRGBoost | $\mathbf{0.625_{\pm0.017}}$ | $\mathbf{0.574_{\pm0.012}}$ | $\underline{0.631_{\pm0.006}}$ | $\mathbf{0.559_{\pm0.003}}$ | $0.993_{\pm0.001}$ | $\mathbf{0.943_{\pm0.003}}$ | $\mathbf{0.724_{\pm0.006}}$ |

## 6 DISCUSSION

While additive tree ensemble models like DEF require no sampling to train and are easy to sample from, we find that, in practice, they require very deep trees to model the data well and still offer subpar performance for sampling and density estimation.

In contrast, NRGBoost was able to model the data better, while using fewer and shallower trees. Its main drawback is that it can only be sampled from approximately using more expensive MCMC and also requires sampling during the training process. Our fast Gibbs sampling implementation, coupled with our proposed amortized sampling approach, were able to mitigate the slow training, making these models more practical. Unfortunately, they remain cumbersome to use for sampling, as autocorrelation between samples from the same Markov chain makes them inefficient in scenarios requiring independent samples. We argue, however, that unlike in image or text generation, where fast sampling is necessary for an interactive user experience, this can be less of a concern when generating synthetic datasets, where the one-time cost of sampling can be less important than faithfully capturing the data-generating distribution.

One advantage of the energy-based approach that we did not explore is that it allows for arbitrary conditional sampling, requiring only that one clamps the values of conditioning variables during Gibbs sampling. In contrast, other methods, such as diffusion-based approaches, are less flexible, since conditioning variables typically need to be predetermined at training time.

## 7 CONCLUSION

We extend the two most popular tree-based discriminative methods for use in generative modeling. We find that our boosting approach, in particular, offers generally good discriminative performance and competitive sampling performance to more specialized alternatives. We hope that these results encourage further research into generative boosting approaches for tabular data, particularly in exploring other applications beyond sampling that are enabled by density models.

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

# Appendix

## A   THEORY

The expected log-likelihood for an energy-based model (EBM),

$$q_f(\mathbf{x}) = \frac{\exp\left(f(\mathbf{x})\right)}{Z[f]}, \tag{13}$$

is given by

$$L[f] = \mathbb{E}_{\mathbf{x}\sim p} \log q_f(\mathbf{x}) = \mathbb{E}_{\mathbf{x}\sim p} f(\mathbf{x}) - \log Z[f]. \tag{14}$$

The *first variation* of $L$ can be computed as

$$\delta L[f; g] \coloneqq \left.\frac{dL[f+\epsilon g]}{d\epsilon}\right|_{\epsilon=0} = \mathbb{E}_{\mathbf{x}\sim p}\, g(\mathbf{x}) - \delta \log Z[f; g] = \mathbb{E}_{\mathbf{x}\sim p}\, g(\mathbf{x}) - \mathbb{E}_{\mathbf{x}\sim q_f}\, g(\mathbf{x}). \tag{15}$$

This is a linear functional of its second argument, $g$, and can be regarded as a directional derivative of $L$ at $f$ along a variation $g$. The last equality comes from the following computation of the first

variation of the log-partition function:

$$\delta \log Z[f; g] = \frac{\delta Z[f; g]}{Z[f]} \tag{16}$$

$$= \frac{1}{Z[f]} \sum_{\mathbf{x}} \exp'\left(f(\mathbf{x})\right) g(\mathbf{x}) \tag{17}$$

$$= \sum_{\mathbf{x}} \frac{\exp\left(f(\mathbf{x})\right)}{Z[f]} g(\mathbf{x}) \tag{18}$$

$$= \mathbb{E}_{\mathbf{x} \sim q_f} g(\mathbf{x}). \tag{19}$$

Analogous to a Hessian, we can differentiate Equation 15 again along a second independent variation $h$ of $f$, yielding a symmetric bilinear functional which we will write as $\delta^2 L[f; g, h]$. Note that the first term in Equation 14 is linear in $f$ and thus has no curvature, so we only have to consider the log-partition function itself:

$$\delta^2 L[f; g, h] := \left. \frac{\partial^2 L[f + \epsilon g + \varepsilon h]}{\partial \epsilon \partial \varepsilon} \right|_{(\epsilon, \varepsilon) = 0} \tag{20}$$

$$= -\delta^2 \log Z[f; g, h] = -\delta \left\{ \delta \log Z[f; g] \right\} [f; h] \tag{21}$$

$$= -\delta \left\{ \frac{1}{Z[f]} \sum_{\mathbf{x}} \exp\left(f(\mathbf{x})\right) g(\mathbf{x}) \right\} [f; h] \tag{22}$$

$$= \frac{\delta Z[f; h]}{Z^2[f]} \sum_{\mathbf{x}} \exp\left(f(\mathbf{x})\right) g(\mathbf{x}) - \frac{1}{Z[f]} \sum_{\mathbf{x}} \exp'\left(f(\mathbf{x})\right) g(\mathbf{x}) h(\mathbf{x}) \tag{23}$$

$$= \frac{\delta Z[f; h]}{Z[f]} \cdot \mathbb{E}_{\mathbf{x} \sim q_f} g(\mathbf{x}) - \frac{1}{Z[f]} \sum_{\mathbf{x}} \exp\left(f(\mathbf{x})\right) g(\mathbf{x}) h(\mathbf{x}) \tag{24}$$

$$= \mathbb{E}_{\mathbf{x} \sim q_f} h(\mathbf{x}) \cdot \mathbb{E}_{\mathbf{x} \sim q_f} g(\mathbf{x}) - \mathbb{E}_{\mathbf{x} \sim q_f} h(\mathbf{x}) g(\mathbf{x}) \tag{25}$$

$$= -\text{Cov}_{\mathbf{x} \sim q_f}\left(g(\mathbf{x}), h(\mathbf{x})\right). \tag{26}$$

Note that this functional is negative semi-definite for all $f$. In other words, we have $\delta^2 L[f; h, h] \leq 0$ for any variation $h$, meaning that the log-likelihood is a concave functional of $f$.

Using these results, we can now compute the Taylor expansion of the increment in log-likelihood $L$ from a change $f \to f + \delta f$ up to second order in $\delta f$:

$$\Delta L_f[\delta f] = \delta L[f; \delta f] + \frac{1}{2} \delta^2 L[f; \delta f, \delta f] \tag{27}$$

$$= \mathbb{E}_{\mathbf{x} \sim p} \delta f(\mathbf{x}) - \mathbb{E}_{\mathbf{x} \sim q_f} \delta f(\mathbf{x}) - \frac{1}{2} \text{Var}_{\mathbf{x} \sim q_f} \delta f(\mathbf{x}). \tag{28}$$

As an aside, defining the functional derivative, $\frac{\delta J[f]}{\delta f(\mathbf{x})}$, of a functional $J$ implicitly by

$$\sum_{\mathbf{x}} \frac{\delta J[f]}{\delta f(\mathbf{x})} g(\mathbf{x}) = \delta J[f; g], \tag{29}$$

we can formally define, by analogy with the parametric case, the Fisher information "matrix" at $f$, as the following bilinear functional of two independent variations $g$ and $h$:

$$F[f; g, h] := -\sum_{\mathbf{y}, \mathbf{z}} \left[ \mathbb{E}_{\mathbf{x} \sim q_f} \frac{\delta^2 \log q_f(\mathbf{x})}{\delta f(\mathbf{y}) \delta f(\mathbf{z})} \right] g(\mathbf{y}) h(\mathbf{z}) \tag{30}$$

$$= \sum_{\mathbf{y}, \mathbf{z}} \frac{\delta^2 \log Z[f]}{\delta f(\mathbf{y}) \delta f(\mathbf{z})} g(\mathbf{y}) h(\mathbf{z}) \tag{31}$$

$$= \delta^2 \log Z[f; g, h] = -\delta^2 L[f; g, h]. \tag{32}$$

To summarize, we have that the Fisher information is the same as the negative Hessian of the log-likelihood for these models. This is essentially due to the fact that the only term in $\log q_f(\mathbf{x})$ that is non-linear in $f$ is the log-partition, which is not a function of $\mathbf{x}$, and therefore it doesn't matter whether the expectation is taken under $p$ or $q_f$.

### A.1 Application to Piecewise Constant Functions

Considering a weak learner such as

$$\delta f(\mathbf{x}) = \sum_{j=1}^{J} w_j \mathbf{1}_{X_j}(\mathbf{x}), \tag{33}$$

where the subsets $X_j$ are disjoint and cover the entire input space, $\mathcal{X}$, we have that

$$\mathbb{E}_{\mathbf{x} \sim q} \delta f(\mathbf{x}) = \sum_{\mathbf{x} \in \mathcal{X}} q(\mathbf{x}) \sum_{j=1}^{J} w_j \mathbf{1}_{X_j}(\mathbf{x}) \tag{34}$$

$$= \sum_{j=1}^{J} w_j \sum_{\mathbf{x} \in X_j} q(\mathbf{x}) = \sum_{j=1}^{J} w_j Q(X_j). \tag{35}$$

Similarly, making use of the fact that $\mathbf{1}_{X_i}(\mathbf{x})\mathbf{1}_{X_j}(\mathbf{x}) = \delta_{ij}\mathbf{1}_{X_i}(\mathbf{x})$, we can compute

$$\mathbb{E}_{\mathbf{x} \sim q} \delta f^2(\mathbf{x}) = \sum_{\mathbf{x} \in \mathcal{X}} q(\mathbf{x}) \left( \sum_{j=1}^{J} w_j \mathbf{1}_{X_j}(\mathbf{x}) \right)^2 = \sum_{j=1}^{J} w_j^2 Q(X_j). \tag{36}$$

In fact, we can extend this to any ordinary function of $\delta f$:

$$\mathbb{E}_{\mathbf{x} \sim q}\, g\left(\delta f(\mathbf{x})\right) = \sum_{\mathbf{x} \in \mathcal{X}} q(\mathbf{x}) \sum_{j=1}^{J} \mathbf{1}_{X_j}(\mathbf{x}) g\left(\delta f(\mathbf{x})\right) \tag{37}$$

$$= \sum_{j=1}^{J} \sum_{\mathbf{x} \in X_j} q(\mathbf{x}) g(w_j) \tag{38}$$

$$= \sum_{j=1}^{J} g(w_j) Q(X_j), \tag{39}$$

where we made use of the fact that the $\mathbf{1}_{X_j}$ constitute a partition of unity:

$$1 = \sum_{j=1}^{J} \mathbf{1}_{X_j}(\mathbf{x}) \qquad \forall \mathbf{x} \in \mathcal{X}. \tag{40}$$

Finally, we can compute the increase in log-likelihood from a step $f \to f + \alpha \cdot \delta f$ as

$$L[f + \alpha \cdot \delta f] - L[f] = \mathbb{E}_{\mathbf{x} \sim p}\left[\alpha \cdot \delta f(\mathbf{x})\right] - \log Z[f + \alpha \cdot \delta f] + \log Z[f] \tag{41}$$

$$= \alpha \mathbb{E}_{\mathbf{x} \sim p} \delta f(\mathbf{x}) - \log \mathbb{E}_{\mathbf{x} \sim q_f} \exp(\alpha \delta f(\mathbf{x})) \tag{42}$$

$$= \alpha \sum_{j=1}^{J} w_j P\left(X_j\right) - \log \sum_{j=1}^{J} Q_f\left(X_j\right) \exp\left(\alpha w_j\right), \tag{43}$$

where in Equation 42 we made use of the equality:

$$\log Z[f + \alpha \cdot \delta f] - \log Z[f] = \log \frac{\sum_{\mathbf{x}} \exp(f(\mathbf{x}) + \alpha \delta f(\mathbf{x}))}{Z[f]} = \log \sum_{\mathbf{x}} q_f(\mathbf{x}) \exp(\alpha \delta f(\mathbf{x})), \tag{44}$$

and of the result in Equation 39 in the final step. This result can be used to conduct a line search over the step-size using training data and to estimate an increase in log-likelihood at each round of boosting, for the purpose of early stopping, using validation data.

## A.2 A Simple Convergence Result

We can take the results from the last subsection to show that each boosting round improves the likelihood of the model under ideal conditions. Substituting the NRGBoost leaf values, $w_j = P(X_j)/Q_f(X_j) - 1$, in Equation 43 we get:

$$L[f + \alpha \cdot \delta f] - L[f] = \alpha \left[ \sum_{j=1}^{J} \frac{P^2(X_j)}{Q_f(X_j)} - 1 \right] - \log \sum_{j=1}^{J} Q_f(X_j) \exp\left[ \alpha \left( \frac{P(X_j)}{Q_f(X_j)} - 1 \right) \right]$$

(45)

$$= \alpha \chi^2(P\|Q_f) - \log \mathbb{E}\exp(\alpha W),$$

(46)

where, with some abuse of notation, we denote by $\chi^2(P\|Q_f)$ the $\chi^2$-divergence between two discrete distributions over $[1..J]$ induced by the partitioning obtained in the current boosting round. This value is always non-negative, and can only be zero if $P(X_j) = Q_f(X_j)$ for all leaves. It therefore corresponds to an increase in likelihood that is as large as one can make the difference between the induced $P$ and $Q_f$ distributions by a judicious choice of partitioning, $X_j$. Note that this quantity is precisely the objective that NRGBoost tries to greedily maximize (see Equation 9).

The second term in Equation 46 can be interpreted as the log of a moment generating function for a centered random variable $W$ that takes values in $\{P(X_j)/Q_f(X_j) - 1\}_{j\in[1..J]}$ (i.e., the set of the leaf values), with probabilities $\{Q_f(X_j)\}_{j\in[1..J]}$. With our proposed regularization approach, we limit the ratio $P(X_j)/Q_f(X_j)$ in our choice of $X_j$ to a maximum value $R$ and therefore we have that $W \in [-1, R-1]$. Given that $W$ is bounded, by Hoeffding's Lemma (1963), it is sub-gaussian with variance proxy $R^2/4$ and the log of its moment generating function is thus upper bounded by:

$$\log \mathbb{E}\exp(\alpha W) \leq \frac{\alpha^2 R^2}{8}.$$

(47)

As a result, we have the following lower bound on the log-likelihood increase:

$$L[f + \alpha \cdot \delta f] - L[f] \geq \alpha \left[ \chi^2(P\|Q_f) - \alpha \frac{R^2}{8} \right].$$

(48)

As long as, in the current round of boosting, a partitioning of the input space can be found that yields a non-zero $\chi^2(P\|Q_f)$, the likelihood is guaranteed to increase as long as we choose a small enough step size, $\alpha < \frac{8}{R^2}\chi^2(P\|Q_f)$. In particular, choosing a step size $\alpha = \frac{4}{R^2}\chi^2(P\|Q_f)$ produces an increase in likelihood of at least $2\left[\chi^2(P\|Q_f)/R\right]^2$.

We note that this result, while insightful, assumes that one uses the exact probabilities under the model distribution in the NRGBoost update. In practice, we would only be able to approximately estimate these with MCMC. We leave further analysis for future work.

## A.3 NRGBoost Algorithm

In Algorithm 1 we provide a high-level overview of the training loop for NRGBoost.

# B Density Estimation Trees and Density Estimation Forests

Density Estimation Trees (DET), introduced by Ram & Gray (2011), model the density function as a piecewise constant function,

$$q(\mathbf{x}) = \sum_{j=1}^{J} v_j \mathbf{1}_{X_j}(\mathbf{x}),$$

(49)

where the $X_j$ are given by a partitioning of the input space, $\mathcal{X}$, induced by a binary tree and the $v_j$ are the density values associated with each leaf that, for the time being, we will only require to be such that $q(\mathbf{x})$ sums to one. Note that it is possible to draw an exact sample from this type of model

---

**Algorithm 1** NRGBoost training

---

**Input:** Empirical training distribution $\hat{p}$, initial model distribution $q_0$, size of the sample pool $M$, fraction of samples to independently drop at each round $p_{refresh} \in [0, 1]$, number of boosting rounds $T$, maximum number of leaves in each tree $J$, maximum probability ratio in each leaf $R > 1$, shrinkage parameter $\gamma \in ]0, 1]$.
**Output:** Energy function $f$.
  $f \leftarrow \log q_0$
  **for** $t \leftarrow 1$ to $T$ **do**
    **if** $t = 1$ **then**                         $\triangleright$ No sampling needed since $q_0$ is normalized
      $\hat{q} \leftarrow q_0$
    **else if** $t = 2$ **then**                      $\triangleright$ Initialize sample pool
      $\hat{q} \leftarrow \{\mathbf{x}_i\}_{i \in [M]} \sim_{iid} \exp f$    $\triangleright$ By rejection sampling using $q_0$ as a proposal distribution
    **else**
      $\hat{q} \leftarrow$ Drop samples from $\hat{q}$ randomly with probability $p_{refresh}$
      $\hat{q} \leftarrow$ Keep each sample, $\mathbf{x}_i$, from $\hat{q}$ with probability given by Equation 12 using $\delta f_{t-1}$
      $\hat{q} \leftarrow$ Add $M - |\hat{q}|$ samples to $\hat{q}$ by Gibbs sampling from $\exp f$
    **end if**
    $\delta f_t \leftarrow$ FitTree$(\hat{p}, \hat{q}; J, R)$    $\triangleright$ Greedily grow tree using Equation 10 as a splitting criterion
    $\alpha_t \leftarrow \arg\max_\alpha \Delta L_f[\alpha \cdot \delta f_t]$         $\triangleright$ Line search on training likelihood (Equation 43)
    $f \leftarrow f + \gamma \cdot \alpha_t \cdot \delta f_t$
  **end for**

---

by randomly selecting a leaf, $j \in [1..J]$, given probabilities $v_j$, and then drawing a sample from a uniform distribution over $X_j$.

To fit a DET, Ram & Gray (2011) proposes optimizing the Integrated Squared Error (ISE) between the data generating distribution, $p(\mathbf{x})$ and the model:

$$\min_{q \in \mathcal{Q}} \sum_{\mathbf{x} \in \mathcal{X}} (p(\mathbf{x}) - q(\mathbf{x}))^2 . \tag{50}$$

Noting that $\bigcup_{j=1}^J X_j = \mathcal{X}$, we can rewrite this as

$$\min_{v_1,\ldots,v_J,X_1,\ldots,X_J} \quad \sum_{\mathbf{x} \in \mathcal{X}} p^2(\mathbf{x}) + \sum_{j=1}^J \sum_{\mathbf{x} \in X_j} \left(v_j^2 - 2v_j p(\mathbf{x})\right)$$
$$\text{s.t.} \quad \sum_{j=1}^J \sum_{\mathbf{x} \in X_j} v_j = 1 . \tag{51}$$

Since the first term in the objective does not depend on the model, this optimization problem can be further simplified as

$$\min_{v_1,\ldots,v_J,X_1,\ldots,X_J} \quad \sum_{j=1}^J \left(v_j^2 V(X_j) - 2v_j P(X_j)\right)$$
$$\text{s.t.} \quad \sum_{j=1}^J v_j V(X_j) = 1 , \tag{52}$$

where $V(X)$ denotes the volume of a subset $X$. Solving this quadratic program for the $v_j$ we obtain the following optimal leaf values and objective:

$$v_j^* = \frac{P(X_j)}{V(X_j)} , \qquad\qquad \text{ISE}^* (X_1, \ldots, X_J) = -\sum_{j=1}^J \frac{P^2(X_j)}{V_f(X_j)} . \tag{53}$$

One can therefore grow a tree by greedily choosing to split a parent leaf, with support $X_P$, into two leaves, with supports $X_L$ and $X_R$, so as to maximize the following criterion:

$$\max_{X_L, X_R \in \mathcal{P}(X_P)} \frac{P^2(X_L)}{V(X_L)} + \frac{P^2(X_R)}{V(X_R)} - \frac{P^2(X_P)}{V(X_P)} . \tag{54}$$

This leads to a similar splitting criterion to Equation 10, but replacing the previous model's distribution with the volume measure $V$, which can be interpreted as the uniform distribution on $\mathcal{X}$ (up to a multiplicative constant).

**Maximum Likelihood** Often, generative models are trained to maximize the log-likelihood of the observed data,

$$\max_{q} \mathbb{E}_{\mathbf{x} \sim p} \log q(\mathbf{x}), \tag{55}$$

rather than the ISE. This was left for future work in Ram & Gray (2011) but, following a similar approach to the above, the optimization problem to solve is:

$$\max_{v_1,\dots,v_J,X_1,\dots,X_J} \quad \sum_{j=1}^{J} P(X_j) \log v_j$$
$$\text{s.t.} \quad \sum_{j=1}^{J} v_j V(X_j) = 1. \tag{56}$$

This is, again, easy to solve for $v_j$ since it is separable over $j$ after removing the constraint using Lagrange multipliers. The optimal leaf values and objective are, in this case:

$$v_j^* = \frac{P(X_j)}{V(X_j)}, \qquad L^*(X_1,\dots,X_J) = \sum_{j=1}^{J} P(X_j) \log \frac{P(X_j)}{V_f(X_j)}. \tag{57}$$

The only change is, therefore, to the splitting criterion which should become:

$$\max_{X_L, X_R \in \mathcal{P}(X_P)} P(X_L) \log \frac{P(X_L)}{V(X_L)} + P(X_R) \log \frac{P(X_R)}{V(X_R)} - P(X_P) \log \frac{P(X_P)}{V(X_P)}. \tag{58}$$

This choice of splitting criterion can be seen as analogous to the choice between Gini impurity and Shannon entropy in the computation of the information gain in decision trees.

**Bagging and Feature Subsampling** Following the common approach in decision trees, Ram & Gray (2011) suggests the use of pruning for regularization of DET models. Practice has, however, evolved to prefer bagging as a form of regularization rather than relying on single decision trees. We employ the same principle for DETs, by fitting many trees on bootstrap samples of the data. We also adopt the common practice from Random Forests of randomly sampling a subset of features to consider when splitting any leaf node, in order to encourage independence between the different trees in the ensemble. The ensemble model, which we call *Density Estimation Forests* (DEF), is thus an additive mixture of DETs with uniform weights, therefore still allowing for normalized density computation and exact sampling.

## C  GREEDY TREE-BASED MULTIPLICATIVE BOOSTING

In multiplicative generative boosting, an unnormalized current density model, $\tilde{q}_{t-1}(\mathbf{x})$, is updated at each boosting round by multiplication with a new factor $\delta q_t^{\alpha_t}(\mathbf{x})$:

$$\tilde{q}_t(\mathbf{x}) = \tilde{q}_{t-1}(\mathbf{x}) \cdot \delta q_t^{\alpha_t}(\mathbf{x}). \tag{59}$$

For our proposed NRGBoost, this factor is chosen in order to maximize a local quadratic approximation of the log-likelihood around $q_{t-1}$, as a functional of the log-density (see Section 3). The motivation behind the greedy approach of Tu (2007) and Grover & Ermon (2018) is to instead make the update factor $\delta q_t(\mathbf{x})$ proportional to the likelihood ratio $r_t(\mathbf{x}) := {}^{p(\mathbf{x})}/q_{t-1}(\mathbf{x})$ directly. Under ideal conditions, this would mean that the method converges immediately when choosing a step size $\alpha_t = 1$. In a more realistic setting, however, this method has been shown to converge under conditions on the performance of the individual $\delta q_t$ as discriminators between real and generated data (Tu, 2007; Grover & Ermon, 2018; Cranko & Nock, 2019).

In principle, this desired $r_t(\mathbf{x})$ could be derived from any binary classifier that is trained to predict a probability of a datapoint being generated (e.g., by training it to minimize a strictly proper loss).

Table 4: Comparison of splitting criterion and leaf weights for the different versions of boosting.

| | Splitting Criterion | Leaf Values (Density) |
|---|---|---|
| Greedy (KL) | $P \log \left( P/Q \right)$ | $P/Q$ |
| Greedy ($\chi^2$) | $P^2/Q$ | $P/Q$ |
| NRGBoost | $P^2/Q$ | $\exp \left( P/Q - 1 \right)$ |

Grover & Ermon (2018), however, proposes relying on the following variational bound of an $f$-divergence to derive an estimator for this ratio:

$$D_f(P\|Q_{t-1}) \geq \sup_{u \in \mathcal{U}_t} \left[ \mathbb{E}_{\mathbf{x} \sim p} \, u(\mathbf{x}) - \mathbb{E}_{\mathbf{x} \sim q_{t-1}} f^*(u(\mathbf{x})) \right] , \tag{60}$$

where $f^*$ denotes the convex conjugate of $f$. This bound is tight, with the optimum being achieved for $u_t^*(\mathbf{x}) = f'(p(\mathbf{x})/q_{t-1}(\mathbf{x}))$ if $\mathcal{U}_t$ contains this function. $(f')^{-1}(u_t^*(\mathbf{x}))$ can thus be interpreted as an approximation of the desired $r_t(\mathbf{x})$.

Adapting this method to use trees as weak learners can be accomplished by considering $\mathcal{U}_t$ in Equation 60 to be defined by tree functions $u = 1/J \sum_{j=1}^{J} w_j \mathbf{1}_{X_j}$, with leaf values $w_j$ and leaf supports $X_j$. At each boosting iteration, a new tree, $u_t^*$, can be grown to greedily optimize the lower bound in the r.h.s. of Equation 60, and setting $\delta q_t(\mathbf{x}) = (f')^{-1}(u_t^*(\mathbf{x}))$. This means that $\delta q_t$ is also given by a tree with the same leaf supports and leaf values $v_j := (f')^{-1}(w_j)$, leading to the following separable optimization problem:

$$\max_{w_1,\ldots,w_J,X_1,\ldots,X_J} \sum_{j}^{J} \left[ P(X_j)w_j - Q(X_j)f^*(w_j) \right] . \tag{61}$$

Note that we drop the iteration indices from this point onward for brevity. Maximizing over $w_j$ with the $X_j$ fixed, we have that $w_j^* = f' \left( P(X_j)/Q(X_j) \right)$, which yields the optimal value

$$J^*(X_1,\ldots,X_j) = \sum_{j} \left[ P(X_j)f' \left( \frac{P(X_j)}{Q(X_j)} \right) - Q(X_j)(f^* \circ f') \left( \frac{P(X_j)}{Q(X_j)} \right) \right] . \tag{62}$$

This, in turn, determines the splitting criterion as a function of the choice of $f$. Finally, the optimal density values for the leaves are given by

$$v_j^* = (f')^{-1}(w_j^*) = \frac{P(X_j)}{Q(X_j)} . \tag{63}$$

It is interesting to note two particular choices of $f$-divergences. For the KL divergence, $f(t) = t \log t$ and $f'(t) = 1 + \log t = (f^*)^{-1}(t)$, which leads to

$$J_{KL}(X_1,\ldots,X_j) = \sum_{j} P(X_j) \log \frac{P(X_j)}{Q(X_j)} \tag{64}$$

as the splitting criterion. The Pearson $\chi^2$ divergence, with $f(t) = (t-1)^2$, leads to the same splitting criterion as NRGBoost. Note, however, that for NRGBoost the leaf values for the multiplicative update of the density are proportional to $\exp \left( P(X_j)/Q(X_j) \right)$ instead of the ratio directly. Table 4 summarizes these results.

Another interesting observation is that a DET model can be interpreted as a single round of greedy multiplicative boosting, starting from a uniform initial model. The choice of the ISE as the criterion to optimize the DET corresponds to the choice of Pearson's $\chi^2$ divergence and log-likelihood to the choice of KL divergence.

## D REPRODUCIBILITY

### D.1 DATASETS

We use 5 datasets from the UCI Machine Learning Repository (Dheeru & Karra Taniskidou, 2017): Abalone, Physicochemical Properties of Protein Tertiary Structure (from hereon referred to as

Protein), Adult, MiniBooNE and Covertype. We also use the California Housing dataset which was downloaded through the Scikit-Learn package Pedregosa et al. (2011) and a downsampled version of the MNIST dataset Deng (2012). Table 5 summarizes the main details of these datasets as well as the number of samples used for train/validation/test for the first cross-validation fold.

Table 5: Dataset Information. We respect the original test sets of each dataset when provided, otherwise we set aside 20% of the original dataset as a test set. 20% of the remaining data is set aside as a validation set used for hyperparameter tuning.

| Abbr | Name | Train + Val | Test | Num | Cat | Target | Cardinality |
|---|---|---|---|---|---|---|---|
| AB | Abalone | 3342 | 835 | 7 | 1 | Num | 29 |
| CH | California Housing | 16512 | 4128 | 8 | 0 | Num | Continuous |
| PR | Protein | 36584 | 9146 | 9 | 0 | Num | Continuous |
| AD | Adult | 32560 | 16280* | 6 | 8 | Cat | 2 |
| MBNE | MiniBooNE | 104051 | 26013 | 50 | 0 | Cat | 2 |
| MNIST | MNIST (downsampled) | 60000 | 10000* | 196 | 0 | Cat | 10 |
| CT | Covertype | 464810 | 116202 | 10 | 2 | Cat | 7 |

* Original test set was respected.

The toy data distribution used in Figure 2 consists of a mixture of eight isotropic Gaussians with standard deviation of 1, placed around a circle with a radius of 8. To obtain a discrete input domain, we discretize the input space in 100 equally spaced bins between -11 and 11.

## D.2 DEF AND NRGBOOST IMPLEMENTATION DETAILS

**Discretization** In our practical implementation of tree based methods, we first discretize the input space by binning continuous numerical variables by quantiles. Furthermore, we also bin discrete numerical variables in order to keep their cardinalities smaller than 256. This can also be interpreted as establishing *a priori* a set of discrete values to consider when splitting on each numerical variable and is done for computational efficiency, being inspired by LightGBM (Ke et al., 2017).

**Categorical Splitting** For splitting on a categorical variable, we once again take inspiration from LightGBM. Rather than relying on one-vs-all splits, we found it better to first order the possible categorical values at a leaf according to a pre-defined sorting function and then choose the optimal many-vs-many split as if the variable was numerical. The function used to sort the values is the leaf value function. For splitting on a categorical variable, $x_i$, we order each possible categorical value $k$ by $\hat{P}(x_i=k, X_{-i})/\hat{Q}(x_i=k, X_{-i})$, where $X_{-i}$ denotes the leaf support over the remaining variables.

**Tree Growth Strategy** We always grow trees in best-first order, meaning that we split the current leaf node that yields the maximum gain in the chosen objective value.

**Line Search** As mentioned in Section 3, we perform a line search to find the optimal step size after each round of boosting in order to maximize the likelihood gain in Equation 43. Because evaluating multiple possible step sizes, $\alpha_t$, is inexpensive, we simply do a grid search over 101 different step sizes, split evenly in log-space over $[10^{-3}, 10]$.

**Code** Our implementation of the proposed tree-based methods is mostly Python code using the NumPy library (Harris et al., 2020) and Numba. We implement the tree evaluation and Gibbs sampling in C, making use of the PCG library (O'Neill, 2014) for random number generation.

## D.3 HYPERPARAMETER TUNING

### D.3.1 XGBOOST

To tune the hyperparameters of XGBoost we use 100 trials of random search with the search space defined in Table 6.

Table 6: XGBoost hyperparameter tuning search space. $\delta(0)$ denotes a point mass distribution at 0.

| Parameter | Distribution or Value |
|---|---|
| learning_rate | LogUniform $\left(\left[10^{-3}, 1.0\right]\right)$ |
| max_leaves | Uniform $(\{16, 32, 64, 128, 256, 512, 1024\})$ |
| min_child_weight | LogUniform $\left(\left[10^{-1}, 10^{3}\right]\right)$ |
| reg_lambda | $0.5 \cdot \delta(0) + 0.5 \cdot \text{LogUniform}\left(\left[10^{-3}, 10\right]\right)$ |
| reg_alpha | $0.5 \cdot \delta(0) + 0.5 \cdot \text{LogUniform}\left(\left[10^{-3}, 10\right]\right)$ |
| max_leaves | 0 (we already limit the number of leaves) |
| grow_policy | lossguide |
| tree_method | hist |

Each model is trained for 1000 boosting rounds on regression and binary classification tasks. For multi-class classification tasks a maximum number of 200 rounds of boosting was used due to the larger size of the datasets and because a separate tree is built at every round for each class. The best model was selected based on the validation set, together with the boosting round where the best performance was attained. The test metrics reported correspond to the performance of the selected model at that boosting round on the test set.

### D.3.2   NGBOOST

To tune the hyperparameters of NGBoost we use 100 trials of random search with the search space defined in Table 7. We use a maximum of 500 rounds of boosting and early stopping with a patience of 50 rounds. For consistency with the hyperparameter tuning setup for the other models, we select the model that achieves the best $R^2$ score in validation.

Table 7: NGBoost hyperparameter tuning search space.

| Parameter | Distribution |
|---|---|
| learning_rate | LogUniform $\left(\left[10^{-3}, 1.0\right]\right)$ |
| max_depth | Uniform $([2..10])$ |
| min_samples_leaf | Uniform $([1..10])$ |

### D.3.3   RFDE

We implement the RFDE method (Wen & Hang, 2022) after quantile discretization of the dataset and therefore split at the midpoint of the discretized dimension instead of the original one. When a leaf support has odd cardinality over the splitting dimension, a random choice is made over the two possible splitting values. Finally, the original paper does not mention how to split over categorical domains. We therefore choose to randomly split the possible categorical values for a leaf evenly as we found that this yielded slightly better results than a random one-vs-all split.

For RFDE models we train a total of 1000 trees. The only hyperparameter that we tune is the maximum number of leaves per tree, for which we test the values $[2^6, 2^7, \ldots, 2^{14}]$. For the Adult dataset, due to limitations of our tree evaluation implementation, we only test values up to $2^{13}$.

### D.3.4   DEF

We train ensembles with 1000 DET models. Only three hyperparameters are tuned, using three nested loops, with each loop running over the possible values of a single parameter in a pre-defined order. These are, in order of outermost to innermost:

1. The maximum number of leaves per tree, for which we test the values $[2^{14}, 2^{12}, 2^{10}, 2^{8}]$. For the Adult dataset we use $2^{13}$ instead of $2^{14}$.

2. The fraction of features to consider when splitting a node. We test the values $[d^{-1/2}, d^{-1/4}, 1]$ with $d$ being the dimension of the dataset.

3. The minimum number of data points per leaf. We test the values $[0, 1, 3, 10, 30]$.

To speed up the hyperparameter tuning process, we exit each loop early when the best value found for the objective does not improve between consecutive iterations. For example, if for $2^{12}$ leaves per tree, and a feature fraction of $d^{-1/2}$, the objective does not improve when going from 3 minimum points per leaf to 10, we move on to the next value of the feature fraction. Similarly, if the best objective found for a feature fraction of $d^{-1/4}$ (when considering possible values of minimum number of points per leaf) does not improve over that for $d^{-1/2}$, we move on to the next value of number of leaves.

### D.3.5 ARF

For ARF we used the official python implementation of the algorithm (Blesch & Wright, 2023). Due to memory (and time) concerns we train models with a maximum of 100 trees for up to 10 adversarial iterations. We use a similar grid-search setup to DEF (with early stopping), tuning only the following two parameters:

1. The maximum number of leaves per tree, for which we test the values $[2^6, 2^7, \ldots, 2^{12}, \infty]$ ($\infty$ denotes unconstrained).

2. The minimum number of examples per leaf, for which we test the values $[3, 5, 10, 30, 50, 100]$.

### D.3.6 NRGBOOST

We train NRGBoost models for a maximum of 200 rounds of boosting. We tune only two parameters, using a similar early stopping setup in the outer loop:

1. The maximum number of leaves, for which we try the values $[2^6, 2^8, 2^{10}, 2^{12}]$ in order. For the CT dataset we also include $2^{14}$ in the values to test.

2. The constant factor by which the optimal step size determined by the line search is shrunk at each round of boosting. This is essentially the *learning rate* parameter. To tune it, we perform a Golden-section search for the log of its value using a total of 6 evaluations. The range we use for this search is $[0.01, 0.5]$.

For regularization we limit the ratio between empirical data density and model data density on each leaf to a maximum of 2. We observed that smaller values of this regularization parameter tend to perform better, but that it otherwise plays a similar role to the shrinkage factor. Therefore, we choose to tune only the latter.

For sampling, we use a sample pool of 80,000 samples for all datasets except for Covertype where we use 320,000. These values are chosen so that the number of samples in the pool are, at minimum, similar to the training set size, which we found to be a good rule of thumb. At each round of boosting, samples are removed from the pool according to our rejection sampling approach (using Equation 12), after independently removing 10% at random. These samples are replaced by samples from the current model using Gibbs sampling.

The starting point of each NRGBoost model was selected as a mixture model between a uniform distribution (10%) and the product of training marginals (90%) on the discretized input space. However, we observed that this mixture coefficient does not have much impact on the results.

### D.3.7 TVAE

We use the implementation of TVAE from the SDV package.[2] To tune its hyperparameters we use 50 trials of random search with the search spaces defined in Table 8.

---

[2] https://github.com/sdv-dev/SDV

Table 8: TVAE hyperparameter tuning search space. We set both `compress_dims` and `decompress_dims` to have the number of layers specified by `num_layers`, with `hidden_dim` hidden units in each layer. We use larger batch sizes and smaller number of epochs for the larger datasets (MBNE, MNIST, CT) since these can take significantly longer to run a single epoch.

| Parameter | Datasets | Distribution or Value |
|---|---|---|
| `epochs` | small | Uniform $([100..500])$ |
| | large | Uniform $([50..200])$ |
| `batch_size` | small | Uniform $(\{100, 200, \ldots, 500\})$ |
| | large | Uniform $(\{500, 1000, \ldots, 2500\})$ |
| `embedding_dim` | all | Uniform $(\{32, 64, 128, 256, 512\})$ |
| `hidden_dim` | all | Uniform $(\{32, 64, 128, 256, 512\})$ |
| `num_layers` | all | Uniform $(\{1, 2, 3\})$ |
| `compress_dims` | all | `(hidden_dim,) * num_layers` |
| `decompress_dims` | all | `(hidden_dim,) * num_layers` |

### D.3.8 TABDDPM

We used the official implementation of TabDDPM,[3] adapted to use our datasets and validation setup. To tune the hyperparameters of TabDDPM we use 50 trials of random search with the same search space used by Kotelnikov et al. (2023).

### D.3.9 FOREST-FLOW

We use the official implementation of Forest-Flow.[4] The only hyperparameters with impact on performance that the Forest-Flow documentation recommends tuning are the number of diffusion timesteps (default of 50) and the number of noise values per sample (default of 100). Due to the fact that Forest-Flow models already take significantly longer to train than the other models, and that increasing either value would lead to slower training, we opted to use these default values.

### D.4 EVALUATION SETUP

**Single-variable inference**  For the single-variable inference evaluation, the best models are selected by their discriminative performance on a validation set. The entire setup is repeated five times with different cross-validation folds and with different seeds for all sources of randomness. For the Adult and MNIST datasets the test set is fixed but training and validation splits are still rotated.

**Sampling**  For the sampling evaluation, we use a single train/validation/test split of the real data (corresponding to the first fold in the previous setup) for training the generative models. The density models used are those previously selected based on their single-variable inference performance on the validation set. For the sampling models (TVAE and TabDDPM) we directly evaluate their ML efficiency using the validation data.

**ML Efficiency**  For each selected model we sample a train and validation sets with the same number of samples as those used in the original data. For NRGBoost we generate these samples by running 64 chains in parallel with 100 steps of burn in and downsampling their outputs by 30 (for the smaller datasets) or 10 (for MBNE, MNIST and CT). For every synthetic dataset, an XGBoost model is trained using the best hyperparameters found on the real data and using a synthetic validation set to select the best stopping round for XGBoost. The setup is repeated 5 times with different datasets being generated for each method and dataset.

**Discriminator Measure**  We create the training, validation and test sets to train an XGBoost model to discriminate between real and generated data using the following process:

---

[3] https://github.com/yandex-research/tab-ddpm
[4] https://github.com/SamsungSAILMontreal/ForestDiffusion

- The original validation set is used as the real part of the training set in order to avoid benefiting generative methods that overfit their training set.

- The original test set is split 20%/80%. The 20% portion is used as the real part of the validation set and the 80% portion as the real part of the test set.

- To form the generated part of the training, validation and test sets for the smaller datasets (AB, CH, PR, AD) we sample data according to the original number of samples in the train, validation and test splits on the real data respectivelly. Note that this makes the ratio of real to synthetic data 1:4 in the training set. This is deliberate, because for these smaller datasets the original validation has few samples and adding extra synthetic data helps the discriminator.

- For the larger datasets (MBNE, MNIST, CT) we generate the same number of synthetic samples as there are real samples on each split, therefore making every ratio 1:1, because the discriminator is typically already too powerful without adding additional synthetic data.

Because, in contrast to the previous metric, having a lower number of effective samples helps rather than hurts, we take extra precautions to not generate correlated data with NRGBoost. We draw each sample by running its own independent chain for 100 steps, starting from an independent sample from the initial model, which is a rather slow process. The setup is repeated 5 times with 5 different sets of generated samples from each method.

## D.5 COMPUTATIONAL RESOURCES

The experiments were run on a Linux machine equipped with an AMD Ryzen 7 7700X 8 core CPU and 32 GB of RAM. The comparisons with TVAE and TabDDPM additionally made use of a GeForce RTX 3060 GPU with 12 GB of VRAM.

## E ADDITIONAL RESULTS

### E.1 STATISTICAL DISTANCE MEASURE

As an additional measure of the statistical dissimilarity between data and model distributions, we evaluate a Wasserstein distance using a similar setup to Jolicoeur-Martineau et al. (2024). Namely, we min-max scale numerical variables, and one-hot encode categorical variables, scaling the latter by 1/2. We use a $L_1$ distance in the formulation of the optimal transport problem, which we solve using the POT python library. Since finding the optimal solution scales cubically with the sample size in the worst case, we sub-sample at most 5,000 samples from the original train or test sets and use an equal number of synthetic samples. We repeat the evaluation for each method and dataset 5 times, using different synthetic data and also different subsampling seeds for the real data (where applicable). Our results for the distance between empirical training and test distribution and synthetic samples from each model are reported in Tables 9 and 10 respectively.

Table 9: Wasserstein distance between empirical train distribution and synthetic samples. Reported results are averages and standard errors over 5 repeated experiments. Smaller is better.

|  | AB | CH | PR | AD | MBNE | MNIST | CT |
|---|---|---|---|---|---|---|---|
| TVAE | 0.299 ±0.005 | 0.195 ±0.010 | 0.274 ±0.011 | 1.075 ±0.024 | **0.366 ±0.055** | 20.59 ±0.041 | 0.808 ±0.029 |
| TabDDPM | 0.467 ±0.021 | **0.153 ±0.007** | **0.190 ±0.005** | **0.895 ±0.019** | 109.6 ±93.14 | - | 0.640 ±0.016 |
| Forest-Flow | 0.234 ±0.020 | 0.192 ±0.007 | 0.238 ±0.009 | 1.239 ±0.024 | 4.214 ±3.683 | 20.73 ±0.574 | 0.896 ±0.022 |
| ARF | **0.199 ±0.015** | 0.233 ±0.009 | 0.232 ±0.007 | 1.363 ±0.018 | 0.698 ±0.684 | 18.14 ±0.060 | 0.703 ±0.019 |
| DEF (KL) | 0.320 ±0.006 | 0.199 ±0.012 | 0.260 ±0.010 | 1.682 ±0.024 | 18.21 ±14.92 | 261.0 ±15.60 | 0.933 ±0.025 |
| NRGBoost | 0.509 ±0.047 | 0.170 ±0.012 | 0.230 ±0.011 | 1.028 ±0.015 | 22.25 ±20.09 | **15.60 ±0.212** | **0.600 ±0.016** |

We found these results to be somewhat sensitive to the choice of normalization of the numericals and categoricals, as well as to the choice of distance in this normalized space. Despite NRGBoost not being as dominant as in the discriminator measure, it still fares well, obtaining the best average rank over all datasets and experiments.

Table 10: Wasserstein distance between empirical test distribution and synthetic samples. Reported results are averages and standard errors over 5 repeated experiments. Smaller is better.

|            | AB             | CH             | PR             | AD             | MBNE           | MNIST          | CT             |
|------------|----------------|----------------|----------------|----------------|----------------|----------------|----------------|
| TVAE       | 0.302 ±0.005   | 0.206 ±0.010   | 0.274 ±0.008   | 1.086 ±0.023   | **0.425 ±0.032** | 20.61 ±0.042   | 0.807 ±0.023   |
| TabDDPM    | 0.500 ±0.023   | **0.169 ±0.009** | **0.195 ±0.003** | **0.938 ±0.029** | 109.5 ±93.13   | -              | 0.646 ±0.022   |
| Forest-Flow | 0.248 ±0.020  | 0.205 ±0.009   | 0.231 ±0.008   | 1.243 ±0.029   | 4.295 ±3.697   | 20.73 ±0.574   | 0.893 ±0.021   |
| ARF        | **0.218 ±0.015** | 0.240 ±0.009 | 0.238 ±0.006   | 1.378 ±0.020   | 0.732 ±0.664   | 18.14 ±0.060   | 0.709 ±0.018   |
| DEF (KL)   | 0.337 ±0.006   | 0.210 ±0.012   | 0.264 ±0.011   | 1.696 ±0.023   | 18.29 ±14.90   | 261.0 ±35.01   | 0.935 ±0.024   |
| NRGBoost   | 0.502 ±0.045   | 0.178 ±0.011   | 0.225 ±0.010   | 1.051 ±0.013   | 22.33 ±20.11   | **15.70 ±0.188** | **0.602 ±0.015** |

Table 11: (C)RPS and MAE results. The reported values are means and standard errors over 5 cross-validation folds. The best method for each dataset is highlighted in **bold** and other methods that are not significantly worse (as determined by a paired $t$-test at a 95% confidence level) are underlined.

|          | AB | | CH | | PR | |
|----------|------------------|------------------|------------------|------------------|------------------|------------------|
|          | RPS ↓            | MAE ↓            | CRPS ↓           | MAE ↓            | CRPS ↓           | MAE ↓            |
| XGBoost  | N/A              | **1.524 ±0.032** | N/A              | 0.290 ±0.009     | N/A              | 2.282 ±0.045     |
| NGBoost  | 1.094 ±0.020     | 1.525 ±0.032 | 0.238 ±0.013  | 0.314 ±0.011     | 1.976 ±0.018     | 2.665 ±0.057     |
| NRGBoost | **1.075 ±0.011** | 1.569 ±0.025     | **0.201 ±0.008** | **0.276 ±0.010** | **1.490 ±0.023** | **2.063 ±0.036** |

## E.2 UNCERTAINTY ESTIMATION IN REGRESSION TASKS

One advantage of density models over discriminative regression models like XGBoost is that they estimate the full conditional distribution, $p(y|\mathbf{x})$, rather than provide a point estimate. In Section 5.1, we focused on a metric that evaluates the point estimate $E[y|\mathbf{x}]$. To further evaluate the quality of the models' probabilistic predictions, in Table 11 we report two other metrics:

- **MAE:** We compute the average Mean Absolute Error given a point estimate of $y$. For NGBoost and NRGBoost, we use the median of the estimated $q(y|\mathbf{x})$ since this is the point estimate that minimizes MAE. Note that, while it is possible to train XGBoost to directly optimize this metric instead of MSE, models estimating $q(y|\mathbf{x})$ are flexible to choose a different point estimate post-training.

- **(C)RPS:** The Continuous Ranked Probability Score is a proper scoring rule commonly used to evaluate a probabilistic forecast (Gneiting & Raftery, 2007). For both NGBoost and NRGBoost, we evaluate this score analytically for the forecasted $q(y|\mathbf{x}_i)$ of every test point. Because the Abalone dataset has discrete numerical targets, we compute RPS for this dataset instead. For this purpose, the continuous distribution obtained with NGBoost is first discretized by rounding to the nearest discrete value within the domain of the target.

Note that we omit the results for ARF from this analysis because the Python implementation does not provide neither conditional nor unconditional density evaluation at the time of writing. For the results reported in Table 1, we implemented our own evaluation of $E[y|\mathbf{x}]$ for these models. Extending this implementation to compute the median or the CRPS analytically is, however, not trivial due to ARF fitting more complex continuous distributions at the leaves than the other tree-based models.

On MAE, we find that NRGBoost outperforms XGBoost convincingly in the California Housing and Protein datasets, in part due to the use of a better point estimate for this metric. The Abalone dataset is the only dataset where predicting the mean of the forecasting distribution would outperform the median for MAE. We also find that NRGBoost achieves a lower (C)RPS than NGBoost on all datasets, which we attribute to the parametric choice made for NGBoost not necessarily being the most well-suited for these datasets. This highlights a key drawback of parametric approaches compared to non-parametric ones: they require domain expertise to avoid suboptimal results.

Table 12: Results for inference with a missing covariate. The original results of each method when having access to the full data are also reported for reference. The best approach for dealing with missing data in each scenario is highlighted in **bold**.

| Model | Imputation | CH ($R^2$ ↑) | AD (AUC ↑) | CT (Accuracy ↑) |
|-------|-----------|--------------|------------|------------------|
| XGBoost | Full Data | $0.849_{\pm0.009}$ | $0.927_{\pm0.000}$ | $0.971_{\pm0.001}$ |
| | Mean | $-0.283_{\pm0.107}$ | N/A | $0.610_{\pm0.004}$ |
| | Median/Mode | $-0.117_{\pm0.107}$ | $0.914_{\pm0.003}$ | $0.621_{\pm0.002}$ |
| | KNN (K=5) | $0.150_{\pm0.107}$ | $0.910_{\pm0.003}$ | $0.883_{\pm0.001}$ |
| NRGBoost | Full Data | $0.850_{\pm0.011}$ | $0.920_{\pm0.001}$ | $0.948_{\pm0.001}$ |
| | Marginalization | $\mathbf{0.773_{\pm0.010}}$ | $\mathbf{0.920_{\pm0.001}}$ | $\mathbf{0.923_{\pm0.001}}$ |

## E.3 INFERENCE WITH MISSING DATA

Generative density models can be used more flexibly for inference than their discriminative counterparts. In this section we present a case study on a problem of inference in the presence of a missing covariate that highlights this advantage.

For each type of task, we choose a dataset and remove an important feature from the respective test set (*Longitude* for the California dataset, *education* for the Adult dataset and *Elevation* for the Covertype dataset). We then attempt to predict the value of the target variable using XGBoost with the help of several imputation techniques that are popular with data science practitioners:

- Mean and median imputation (for a missing numerical feature).

- Mode imputation (for a missing categorical feature).

- Imputation with the mean or mode of the 5 nearest neighbors on the training data. Note that this imputation requires one to have access to the entire training set at test time.

NRGBoost can handle this type of problem in a principled manner by marginalizing over the possible values of the missing feature, rather than relying on *ad-hoc* imputation. Our results, in Table 12, show that this approach is able to outperform imputation, even on datasets where a performance gap between NRGBoost and XGBoost existed on the full data.

## E.4 OTHER CHALLENGING TASKS

In order to test the limits of NRGBoost, we compare its performance to XGBoost on the following two challenging tasks taken from Gorishniy et al. (2021):

- **ALOI:** This is an image classification dataset with 128 features and 100k examples distributed evenly over 1000 classes. Because our implementation of NRGBoost is limited to variables with a cardinality of 255, we consider only data for the first 250 classes. This still constitutes 25 times more classes than MNIST, presenting a challenge to both discriminative and generative models.

- **Microsoft:** This regression task contains both a large number of samples (1.2M) and features (136), making it one order of magnitude larger than MNIST and Covertype in terms of the product of the two. Because of this, we found that our usual hyperparameter tuning protocol for NRGBoost to be impractical and had to resort to manual tuning instead. The best model we found had 256 leaves and was trained for 500 boosting iterations with a maximum ratio in each leaf of 4 and a shrinkage factor of 0.1.

We use the same train, validation and test splits used in Gorishniy et al. (2021) and report the same metrics in Table 13. We find that NRGBoost outperforms XGBoost in the ALOI dataset but that there is still a significant gap between the best NRGBoost model that we were able to find on the Microsoft dataset and XGBoost.

Table 13: Comparison of XGBoost and NRGBoost on ALOI and Microsoft datasets.

|  | **ALOI** (Accuracy ↑) | **Microsoft** (RMSE ↓) |
|---|---|---|
| XGBoost | 0.943 | 0.744 |
| NRGBoost | 0.962 | 0.787 |

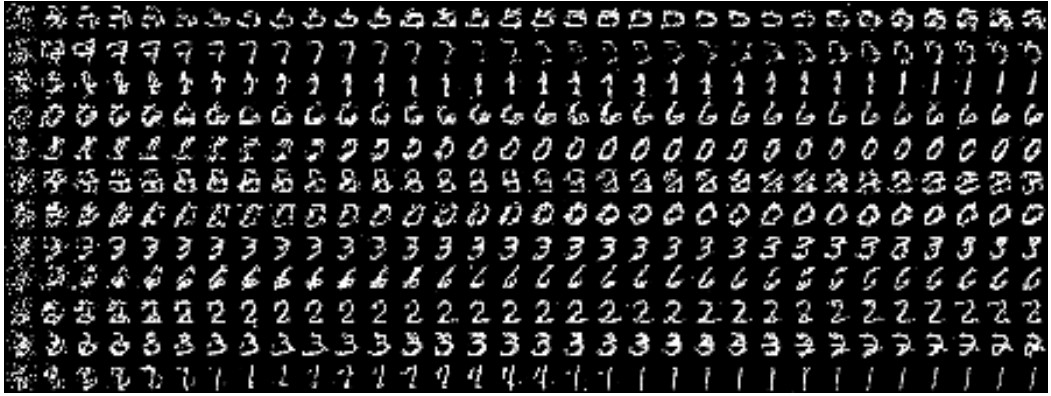

Figure 5: Downsampled MNIST samples generated by Gibbs sampling from an NRGBoost model. Each row corresponds to an independent chain initialized with a sample from the initial model $f_0$ (first column). Each column represents a consecutive sample from the chain.

### E.5 MCMC Chain Convergence on MNIST

Figure 5 shows the convergence of Gibbs sampling from an NRGBoost model. In only a few samples each chain appears to have converged to the data manifold after starting at a random sample from the initial model (a mixture between the product of training marginals and a uniform). Note how consecutive samples are autocorrelated. In particular it can be rare for a chain to switch between two different modes of the distribution (e.g., switching digits) even though a few such transitions can be observed.

### E.6 Computational Effort

#### E.6.1 Training

In Figure 6 we report the training times for NRGBoost, as well as the other methods, for the best model selected by hyperparameter tuning. We do not report the training times for the RFDE method because it is virtually free when compared to the other methods given that the splitting process is random and does not depend on the data, only on the input domain.

Note that the biggest computational cost for training an NRGBoost model is the Gibbs sampling, accounting for roughly 70% of the training time on average. This could potentially be improved by leveraging higher parallelism than what we used (16 virtual cores).

We note also that we believe that there is still plenty of margin for optimizing the tree-fitting code used for DEF models. As such, the results presented here are merely indicative.

The computational effort required for fitting a tree-based model should scale linearly with both the number of samples and the number of features. This is the main source of variation for the training times across datasets. However, we note that larger datasets can benefit more from larger models (e.g., with a larger number of leaves) which also take a longer time to train.

#### E.6.2 Sampling

In Table 14, we report the time required to draw 10,000 independent samples from each model on each dataset. We do not report sampling times for DEF models because our current implementation

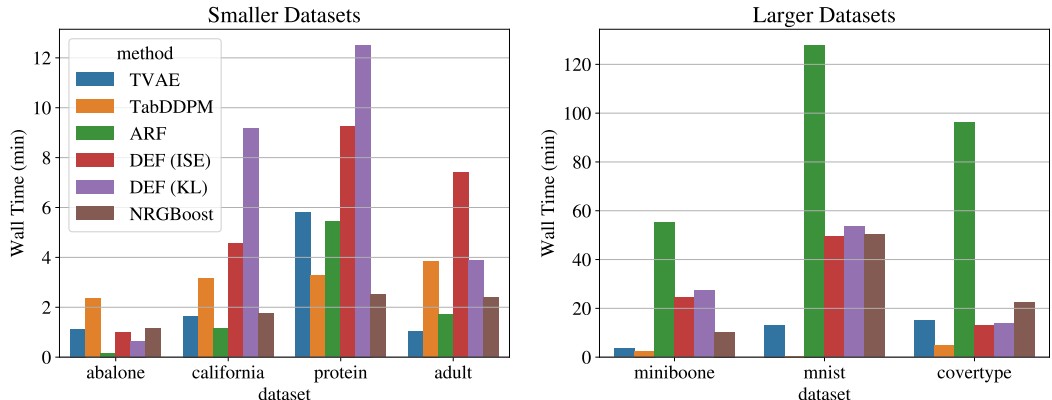

Figure 6: Wall time required to train the best model for each method.

Table 14: Time required to draw 10000 independent samples in seconds. The last columns shows how much faster a method is compared to NRGBoost in the median case. We find that TabDDPM, the only method that realistically competes with NRGBoost in sample quality is, in the median case, 10x faster for sampling.

|  | AB | CH | PR | AD | MBNE | MNIST | CT | $\times$ Faster (Median) |
|---|---|---|---|---|---|---|---|---|
| TVAE | 0.06 | 0.05 | 0.04 | 0.05 | 0.22 | 1.33 | 0.07 | 984.0 |
| TabDDPM | 42.3 | 7.36 | 6.82 | 1.68 | 1.13 | - | 9.69 | 9.67 |
| ARF | 0.38 | 0.22 | 0.40 | 2.62 | 3.84 | - | 4.47 | 55.7 |
| NRGBoost | 23.1 | 49.2 | 54.6 | 65.0 | 194.2 | 622.4 | 109.9 | - |

of this procedure is very suboptimal, taking even longer than sampling from NRGBoost. We believe, however, that a more optimized approach would, at most, be somewhat slower than sampling from ARF models, since the DEF ensembles are larger.

For NRGBoost, independent samples are generated by starting from an independent sample from $q_0$ and then running a fixed number, $B$, of Gibbs sampling iterations, taking only the last iterate as a sample. However, this process only produces an approximate sample from the energy model, as it remains biased by the initial sample, which is drawn from an incorrect distribution. The larger $B$ is, compared to the Markov chain's mixing time, the more independent the final sample is from the initial one and the smaller this bias. This parameter, therefore, trades-off correctness for computational time, which scales linearly with $B$.

In our experiments we always set $B = 100$, but this may be excessive for some datasets. We find that $B = 30$ already leads to similar ML efficiency results on most datasets, while cutting down sampling times by a factor of three. Note also that the process of drawing samples is embarrassingly parallel. The numbers we report correspond to running 16 chains in parallel on a CPU with 16 virtual cores. If we were to leverage a CPU with higher core count, we could similarly cut down on sampling times. In fact, the process of sampling from an EBM is not dissimilar to the process of sampling from a diffusion model like TabDDPM, but the latter can leverage the massive parallelism afforded by GPUs.

