# OpenReview forum: "NRGBoost: Energy-Based Generative Boosted Trees"
_ICLR.cc/2025/Conference — ICLR 2025 Poster_

### Official Review · Reviewer_NED1 · 2024-10-17

**Soundness:** 2
**Presentation:** 2
**Contribution:** 4
**Rating:** 10
**Confidence:** 4

**Summary:**

NRGBoost is an energy-based method for tabular data density estimation and generation using gradient boosted decision trees (GBDTs), in contrast to recent diffusion-based generative methods which cannot give density estimates.

**Strengths:**

1. The method is original and well-motivated, and the approach for incorporating rejection sampling within the boosting process to improve efficiency was an excellent contribution towards making energy-based tabular modeling practical.

2. The paper is well-written, with the method and experiments clearly explained. Showing how the model progressively models the dataset over boosting iterations in Figure 2 gave helpful insight into how the method works.

3. The experiments on downsampled MNIST data were very nice to see, and convincing that NRGBoost goes beyond current SotA (ForestFlow) [2] in generation in certain settings, besides also offering density estimation.

**Weaknesses:**

1. [UPDATE: fully addressed in rebuttal] For single-variable inference in Section 6.1, using R^2 as an evaluation metric seems strange, compared to using CRPS, RMSE, and MAE, as employed in [3]. Furthermore, the choice of baseline methods seems incomplete. For example, because Forest-Flow supports conditioning on covariates, one could also generate (say) 100 Forest-Flow conditional samples, then compute the mean and evaluate this.

2. [UPDATE: mostly addressed in rebuttal] The tabular datasets used for evaluating NRGBoost in Section 6.2 were nonstandard. Most recent papers either evaluate on the TabCSDI [1] benchmark of 6 datasets or the ForestFlow [2] benchmark of 27 datasets. Because using one's own choice of datasets does open the possibility for cherry-picking, one should either use a preexisting benchmark or justify one's selection of datasets.

3. [UPDATE: fully addressed in rebuttal] Some relevant works are missing from related work (and perhaps the experiments). It would be good to discuss how this compares against a previous energy-based modeling method, TabEBM [4], albeit one which uses TabPFN instead of GBDTs. And there is UnmaskingTrees [5] which uses GBDTs to perform both generation and also density estimation via autoregression and recursive partitioning.

[1] Zheng, S., & Charoenphakdee, N. (2022). Diffusion models for missing value imputation in tabular data. arXiv preprint arXiv:2210.17128.

[2] Jolicoeur-Martineau, A., Fatras, K., & Kachman, T. (2023). Generating and Imputing Tabular Data via Diffusion and Flow-based Gradient-Boosted Trees. arXiv preprint arXiv:2309.09968.

[3] Beltran-Velez, N., Grande, A. A., Nazaret, A., Kucukelbir, A., & Blei, D. (2024). Treeffuser: Probabilistic Predictions via Conditional Diffusions with Gradient-Boosted Trees. arXiv preprint arXiv:2406.07658.

[4] Margeloiu, A., Jiang, X., Simidjievski, N., & Jamnik, M. (2024). TabEBM: A Tabular Data Augmentation Method with Distinct Class-Specific Energy-Based Models. arXiv preprint arXiv:2409.16118.

[5] McCarter, C. (2024). Unmasking Trees for Tabular Data. arXiv preprint arXiv:2407.05593.

**Questions:**

Questions

- The methods section and experiments devoted to DEF seems out of place, given that it is seemingly unrelated to NRGBoost and given that it performs worse. It is nice to tell more people about DETs and how they can be applied and improved, but I suggest perhaps omitting it entirely and submitting it as an ICLR Blogpost or similar venue.

- I realize that NGBoost does not support nonparametric probabilistic prediction, but I think seeing its results in Table 1 would still be helpful.

Minor suggestions for improved readability:

Replace: We therefore estimate these quantities, with recourse to empirical data for P(X), and to samples approximately drawn from the model with MCMC.
With: We therefore estimate these quantities, with recourse to empirical data for P(X) and to samples approximately drawn from the model with MCMC.

Replace: Because even if the input space is not partially discrete, f is still discontinuous and constant almost everywhere we can’t use gradient based samplers and therefore rely on Gibbs sampling instead.
With: Because, even if the input space is not partially discrete, f is still discontinuous and constant almost everywhere, we can’t use gradient based samplers; we therefore rely on Gibbs sampling instead.

---

> ### Author Response · Authors · 2024-11-21
> **Author Response (Part 1/2)**
>
> Thank you for your detailed feedback and helpful suggestions. Below we address your questions and concerns.
>
> > **Regression Task metrics and NGBoost**
>
> We have added, as requested, a NGBoost discriminative baseline for the regression datasets in Table 1.
>
> We also agree that it would be interesting to use of a proper scoring rule like CRPS to evaluate the quality of the estimated $p(y|x)$ instead of the quality of a point estimate derived from it.
> Our choice of $R^2$ was, for the most part, due to practical reasons. Namely, it allows us to use XGBoost as a baseline, share the same metric between Table 1 and 2 and align better the hyperparameter tuning objectives between density models and sampling models.
> Note that $R^2$ is just a rescaled version of MSE and thus comparing $R^2$ is equivalent to comparing RMSE on a fixed dataset.
> We prefer the former simply for the consistency of its range and because it makes it immediately apparent if the model is better than predicting the average value.
>
> We have additionally added a section in **Appendix E.2** of the revised manuscript, reporting MAE and CRPS for the models in Table 1, together with a brief discussion on uncertainty estimation.
> We find that NRGBoost outperformed NGBoost in CRPS on all 3 regression datasets.
>
> > **Forest-Flow as a Density Baseline**
>
> The idea of conditional sampling to obtain empirical estimates of $p(y|x)$ with Forest-Flow is an interesting suggestion.
> However, as far as we understand, this would require training a specialized Forest-Flow model that only learns the density over $y$ conditioned on every other variable rather than use the model we already trained for unconditional sampling in Section 5.2.
> Such a model would essentially be a discriminative version of Forest-Flow, similar to Treeffuser, and not really useful as a generative model since it can't sample over the conditioning variables.
> Given that our main goal is to compare generative models, we don't believe we would benefit from including yet another discriminative baseline, particularly
> one for which inference is not as seamless and fast as for all the other methods we compare in Table 1.
>
> > **Choice of Datasets**
>
> NRGBoost has been ongoing work for a few years at this point and we have added both datasets and baselines over this period.
> Originally, the main works we compared to ([1][2]) used different datasets.
> We tried, for the most part, to select datasets from these works that span a diverse range of tasks and dataset sizes,
> avoiding tasks that seemed too easy (where logistic regression already achieves a nearly perfect score for example)
> or redundant (e.g., using both adult and census which have the same task and are derived from the same source data).
>
> We find that, for the most part, recent works propose their own set of datasets (e.g., [3][4]) and we fail to see a convergence to any standard one.
> The benchmark that was recently proposed in [5] comprises only relatively small datasets (around the size of the Abalone dataset).
> We are therefore concerned that it is too focused on data scarce scenarios.
>
> However, we acknowledge that cherry-picking is a valid concern.
> We have included some extra results on two challenging tasks from [6] at the request of reviewer `cBTV` but we are open to use the remaining time for the rebuttal to add a full set of datasets from a previous work if the reviewer feels strongly that this effort is necessary.
>
> [[1]](https://arxiv.org/abs/1907.00503) Lei Xu et al. Modeling Tabular data using Conditional GAN. In NeurIPS 2019.
>
> [[2]](https://arxiv.org/abs/2209.15421) Akim Kotelnikov et al. TabDDPM: Modelling Tabular Data with Diffusion Models. In ICML 2023.
>
> [[3]](https://arxiv.org/abs/2210.04018) Jayoung Kim et al. STaSy: Score-based Tabular data Synthesis. In ICLR 2023.
>
> [[4]](https://arxiv.org/abs/2310.09656) Hengrui Zhang et al. Mixed-Type Tabular Data Synthesis with Score-based Diffusion in Latent Space. In ICLR 2024.
>
> [[5]](https://arxiv.org/abs/2309.09968) Alexia Jolicoeur-Martineau et al. Generating and Imputing Tabular Data via Diffusion and Flow-based Gradient-Boosted Trees. In AISTATS 2024.
>
> [[6]](https://arxiv.org/abs/2106.11959) Yury Gorishniy et al. Revisiting Deep Learning Models for Tabular Data. In NeurIPS 2021.

---

> ### Author Response · Authors · 2024-11-21
> **Author Response (Part 2/2)**
>
> > **Inclusion of DEF models**
>
> We agree that the improvements to DETs are not the main contribution and the section dedicated to it is somewhat out of place.
> This contribution dates to a time when other tree-based generative models like ARF and Forest-Flow were not yet available for comparison.
> We decided to include it because:
> - They are the generative analogue of a Random Forest, thus fitting the general story
> - DEF models can outperform some of the other baselines (TVAE/ARF/ForestFlow)
>
> We are open to excluding DEF models from the manuscript but are wary of making such a drastic modification at this stage without the opinion of the other reviewers since it was one of the originally stated contributions of our work.
> As a compromise we have, for the time being, moved the contents of the DEF section fully into Appendix B and added a discussion on DET models to the related work section instead, keeping the DEF results in the experiments section. We believe that this already improves the flow and focus of the paper.
>
>
> > **Missing related work**
>
> We added a discussion to our related work section on both of these recent methods (highlighted in blue). Below we share our thoughts on how each approach compares to NRGBoost:
>
> - **TabEBM**
> relies on an auxiliary classification task to discriminate data from each class from "negative" outlier data, reinterpreting the classifier logits as class-conditional energy functions.
> This avoids maximum likelihood training and associated MCMC sampling. While the approach is somewhat ad-hoc it was shown to work well empirically.
> It requires, however, the existance of a categorical *target* variable to condition on.
> Furthermore, it's reliance on TabPFN as a classifier could prevent its application to larger datasets
> but it would be interesting to know if the same approach would work well with a different classifier.
>
> - **Unmasking Trees**
>  is a sensible autoregressive approach that should, by nature, do well on our inference task since it essentially trains an XGBoost based predictor for each feature conditioned on an arbitrary number of covariates.
> In contrast, NRGBoost trains a single ensemble of trees which can be used directly to predict any feature rather than one predictor per feature.
> The main apparent downside with Unmasking Trees is the need for replicating the training data multiple times (in the paper 50 times the number of dimensions), to expose each predictor to different missing data patterns.
> This would seem to make computational time for training these models scale cubically in the number of features thus preventing application to higher dimentional datasets.
>
> We hope our response adequately addresses your main concerns and welcome any further discussion.

---

> > ### Comment · Reviewer_NED1 · 2024-11-21
> > **Re: Author Response**
> >
> > Thanks for your thorough responses. While I would still like to see results for [5]'s benchmark (even though it targets more-scarce scenarios than this method is really focused on), the rebuttal mostly addressed my concerns. I therefore increased my score from 6 to 8.
> >
> > I concur with the authors' evaluation of Review a4Ct as being of poor quality. Many such cases!

---

> > > ### Author Response · Authors · 2024-11-22
> > > **Thank you**
> > >
> > > Thank you again for your thoughtful feedback and for raising your score. We are glad we were able to address most of your concerns.

---

> > > ### Author Response · Authors · 2024-11-29
> > > **Results on more datasets**
> > >
> > > We did not want to leave any doubts in the reviewers' minds about our choice of datasets so we have made an effort to rerun our experiments on the full set of datasets from [1] in order to show that the results we report generalize to other datasets.  We selected this set of datasets because they are larger but not as numerous as those in [2] (which includes 27 datasets). This makes it more manageable to run and curate results according to our previous setup (same hyperparameter tuning and cross-validation strategy).
> > > Furthermore, these datasets were also recently used in the concurrent work [3].
> > > We will make an effort, in the future, to also run NRGBoost on the setup from [2] (with no hyperparameter tuning).
> > >
> > > We have made one adjustment: to log-transform the target variable of the News dataset which represents the number of times a news article was shared and can span multiple orders of magnitude.
> > > RMSE on this dataset was dominated by errors on news articles that "went viral" and XGBoost would barely get a $R^2$ above 0.
> > > We believe it is more sensible to predict the log of the number of shares in this scenario.
> > > Note that with this change we do not observe TabDDPM "collapsing" on this dataset as reported in [1].
> > > We focused only on the baselines that were most challenging to NRGBoost: ARF as a density model and TabDDPM for sampling.
> > >
> > > > Inference (AUC for classification, $R^2$ for regression, higher is better)
> > >
> > > |          | Default   | Shoppers  | Magic     | Beijing   | News      |
> > > |----------|-----------|-----------|-----------|-----------|-----------|
> > > | XGBoost  | 0.783     | 0.932     | 0.935     | 0.857     | 0.169     |
> > > | ARF      | 0.759     | 0.903     | **0.934** |-$^\dagger$| 0.106     |
> > > | NRGBoost | **0.769** | **0.927** | 0.933     | **0.815** | **0.115** |
> > >
> > > $^\dagger$: Could not get a reasonable density model
> > >
> > > > ML Efficiency (AUC for classification, $R^2$ for regression, higher is better)
> > >
> > > |          | Default   | Shoppers  | Magic     | Beijing   | News      |
> > > |----------|-----------|-----------|-----------|-----------|-----------|
> > > | TabDDPM  | **0.781** | 0.927     | **0.924** | 0.665     | 0.085     |
> > > | ARF      | 0.764     | 0.888     | 0.908     | 0.575     | 0.111     |
> > > | NRGBoost | 0.775     | **0.928** | 0.917     | **0.689** | **0.131** |
> > >
> > > > Sample Discrimination (AUC for classifying real/synthetic samples, lower is better)
> > >
> > > |          | Default   | Shoppers  | Magic     | Beijing   | News      |
> > > |----------|-----------|-----------|-----------|-----------|-----------|
> > > | TabDDPM  | 0.894     | **0.766** | 0.598     | 0.631     | **0.997** |
> > > | ARF      | 0.999     | 0.999     | 0.837     | 0.994     | 1.0       |
> > > | NRGBoost | **0.844** | 0.997     | **0.551** | **0.578** | 1.0       |
> > >
> > > We believe that these results are inline with those reported in our original datasets:
> > > - Small gap between XGBoost and NRGBoost (larger on News)
> > > - NRGBoost (mostly) outperforms ARF in inference (in the Magic dataset all methods achieve very similar results within the noise level)
> > > - ML Efficiency and sample discrimination remain contested between TabDDPM and NRGBoost.
> > >
> > >
> > >
> > > [[1]](https://arxiv.org/abs/2310.09656) Hengrui Zhang et al. Mixed-Type Tabular Data Synthesis with Score-based Diffusion in Latent Space. In ICLR 2024.
> > >
> > > [[2]](https://arxiv.org/abs/2309.09968) Alexia Jolicoeur-Martineau et al. Generating and Imputing Tabular Data via Diffusion and Flow-based Gradient-Boosted Trees. In AISTATS 2024.
> > >
> > > [[3]](https://arxiv.org/abs/2410.20626) Juntong Shi et al. TabDiff: a Multi-Modal Diffusion Model for Tabular Data Generation. ArXiv preprint 2024.

---

> > > > ### Comment · Reviewer_NED1 · 2024-11-30
> > > > **Thanks for the extra experiments!**
> > > >
> > > > Thanks for the extra experiments!

---

### Official Review · Reviewer_a4Ct · 2024-10-31

**Soundness:** 3
**Presentation:** 2
**Contribution:** 2
**Rating:** 6
**Confidence:** 3

**Summary:**

The paper proposes a tree-based approach to model tabular data density which they claim outperforms traditional generative approaches.

**Strengths:**

The paper is confusing and hard to follow. I was unable to find any strength after investing a long time.

**Weaknesses:**

The paper is poorly written, incoherent and uses inconsistent experiments.

**Questions:**

1. The paper is hard to follow and haphazardly written. It uses some vague terms (mid-range consumer CPU, Line 88 and the best generative models, Line 89). Minor details Line 41: please avoid abbreviation like don't, can't.

2. The paper proposes generative model for tabular data in the abstract, but conducts experiments with vision data like MNIST. I am not sure how downsampling makes MNISt relevant to their setting. I did not understand what is going on in Figure 2. There is no description of the data generation process or distribution.

3. The equations are incoherent and sometimes most probably wrong. For example, I do not see the purpose of Equation 2. It was never used later. The math is incoherent and written in such a way that it is hard for me to evaluate their correctness.

4. Table 1: the authors say they bold the best performing algorithms, but they kept their algorithm bold always although it did not perform the best.

5. The paper has bad structure. Related works comes at Section 5 near the end of the paper. There is no flow diagram or pseudocode for their algorithm.

Overall, I think the paper is far from being a coherent and legit study.

---

> ### Author Response · Authors · 2024-11-21
> **Author Response**
>
> Thank you for your time and feedback. We address your questions below.
>
> **Q1: Clarity**
>
> Your feedback regarding clarity stands in contrast to that of the other reviewers, all of whom mentioned the clarity of our exposition as a key strength of the paper.
> We have fixed the colloquialisms and abbreviations in our revised manuscript. Thank you for pointing these out.
>
> **Q2: MNIST experiments**
>
> As mentioned in the paper, this dataset allows us to visually inspect samples from generative models, something that is generally impossible with tabular data.
> Furthermore, two of the main works ([1][2]) we compare to also include experiments on this dataset.
> We will also note that other reviewers (`NED1`) praised the inclusion of these experiments and found them valuable.
>
> [[1]](https://arxiv.org/abs/1907.00503) Lei Xu et al. Modeling Tabular data using Conditional GAN. In NeurIPS 2019.
>
> [[2]](https://arxiv.org/abs/2205.09435) David S. Watson et al. Adversarial random forests for density estimation and generative modeling. In AISTATS 2023.
>
> **Q2: Figure 2**
>
> We have improved the caption of this figure to provide additional details on the data distribution.
> Please note that this Figure is only meant to give a visual intuition of how NRGBoost converges to a toy data distribution by iteratively adding tree-based weak learners.
> It was never meant to be a proper experiment as we are not really comparing to anything.
>
> **Q3: Correctness**
>
> Again, your feedback directly contradicts that of the other reviewers who found our derivations and proofs solid and well explained.
>
> We also find comments such as
> > The equations are incoherent and sometimes most probably wrong.
>
> to be unprofessional and not actionable.
> The only example the reviewer provided, Equation 2, is in the background section on Energy-Based Models to explain how one can perform inference from EBMs despite not knowing the normalizing constant.
> It can be trivially derived and is there merely to give additional background context on Energy-Based models as, we believe, should be evident by the paragraph in which it is inserted.
> Furthermore, it is this equation that we use to compute the results in Table 1.
>
> **Q4: Use of bold in Table 1**
>
> As mentioned in the **caption** to Table 1:
> > The best *generative* method for each dataset is highlighted in **bold** (...)
>
> The only method beating NRGBoost in Table 1 is XGBoost which is not a *generative* method. As mentioned in the text, this baseline is only there to provide an upper bound on the performance that can reasonably be achievable in this task:
>
> > We use XGBoost (Chen & Guestrin, 2016) as a baseline for what should be achievable by a strong discriminative model.
>
> We point out that the main goal of this table is to compare generative methods among themselves, not to discriminative ones.
>
> **Q5: Positioning of the Related Work Section**
>
> We find that most papers nowadays have the Related Work section at the end, often together with the discussion.
> This allows one to better contextualize prior work and how it relates to one's proposed method, after the method is explained to the reader.
> We find that in our particular instance this section works better before the experiments section since it provides some helpful context on some of the baselines we compare against.
>
> **Summary**
>
> We feel that this review was approached from a particularly critical and uncharitable perspective.
> We kindly request the reviewer to reconsider their stance in light of the positive feedback provided by the other reviewers and to give our work a fair and balanced evaluation.

---

> ### Comment · Reviewer_a4Ct · 2024-11-21
> **Comments**
>
> I have several comments:
>
> - Clarity: I stand by my comment on the clarity of the paper as many of the visualizations (Figure 1, 2) and results (discriminative performance) do not make sense to me.
>
> - MNIST: Vision datasets differ significantly from tabular datasets. In images, each pixel is defined by three attributes: its $(x, y)$ coordinates and its magnitude value. In contrast, tabular data is typically represented as feature vectors without spatial relationships. This distinction is a key reason why convolutional neural networks (CNNs) perform well on vision data—they effectively capture spatial information by considering both individual pixel attributes and local neighborhood patterns.
> The references provided by the author only consider the flattened version of the MNIST dataset (see Section 5.1 of Lei et al.) and do not visualize the data in its original image form. To reconstruct the MNIST dataset as an image, each pixel must be arranged based on its $(x, y)$ coordinates, and the magnitude values must represent the true structure of the image. This contrasts with the flattened representation used for the models in the referenced studies, which discards the spatial relationships intrinsic to image data.
>
> - Figure 2: What is the sample size here? What is the radius of the circle? Why did the author choose a particular set of hyper parameters? What happens if they choose shallower or deeper trees?
>
> - Correctness: $q_f$ is absent in the right hand side of Equation 3 and how does $q_f$ appear in the right hand side of Equation 4? The steps taken to derive Equation 4 would clarify the math. I do not understand this line, "We note that just like the original log-likelihood, this Taylor expansion is invariant to adding an overall constant to $\delta f$. This means that, in maximizing equation 4 we can consider only functions that have zero expectation under $q_f$."
> Why maximizing Equation 5 will improve the current energy function $f_t$ and how is $f_t$ defined and what is $t$?
> How the size of $H_t$ in Equation 6 impact their solution and how are they choosing a relatively small constrained set $H_t$?
>
> - Table 1: It is confusing as the author put both discriminative and generative models in the same table where the discriminative one performs better and still bolds a particular set of the algorithms. As I am reading this comment, I am even more confused about how the author performed discrimination between class labels. They estimated an unnormalized density and nowhere in the paper, it is mentioned how they convert their class conditional density into posteriors $p(y|x)$ and thereby, do the discrimination. At the same time, the above point proves my concern about the organization and completeness of the equations.
>
> - Related work section: I disagree with the author that putting related work section at the end makes the paper coherent. Note that they compared a bunch of baselines with their approach without describing their relevance in the relevant work section prior to doing the experiments. I see the authors moved related works to Section 4 from Section 5, but did not mention it in their rebuttal. I also do not see any pseudocode or flow diagram explaining the algorithm yet.

---

> > ### Author Response · Authors · 2024-11-22
> > **Response (Part 1/2)**
> >
> > We address your comments below:
> >
> > > **MNIST**
> >
> > We are treating MNIST as a tabular dataset in the same way as previous works and we don't understand what could have lead you to believe otherwise.
> > For clarity, as far as our method is concerned, it is learning a density over a 197 dimensional input space consisting of a flattened image and a label.
> > We could have chosen a random fixed permutation of the pixels and the result would have been the same.
> > The models we train have no notion of any spatial separation between pixels, which, as you point out, is what makes this problem challenging for tabular models.
> >
> > For all methods, the visualizations in Figure 2 are generated in the same way: we draw a 197 dimensional sample from the model and reshape the image part back to 14x14.
> > Visualization of these samples still lets one qualitatively assess if they make sense and are similar to real samples.
> > Note that [1] shows the same visualizations in the Appendix.
> >
> > [1] David S. Watson et al. Adversarial random forests for density estimation and generative modeling. In AISTATS 2023.
> >
> > > $q_f$ is absent in the right hand side of Equation 3 and how does $q_f$ appear in the right hand side of Equation 4?
> >
> > The $q_f$ appearing in Equation 3 is just shorthand for the mapping from an energy function $f$ to a probability measure given by Equation 1.
> > That is, we take $f$ as the actual argument of the functional on the left hand side.
> >
> > > The steps taken to derive Equation 4 would clarify the math
> >
> > The steps taken to derive Equation 4 are at the beginning of Appendix A.
> >
> > > I do not understand this line, "We note that just like the original log-likelihood, this Taylor expansion is invariant to adding an overall constant to $\delta f$. This means that, in maximizing equation 4 we can consider only functions that have zero expectation under $q_f$."
> >
> > This means that the log-likelihood functional for an EBM has the property that if $c$ is a constant function then:
> >
> > $$L[f + c] = L[f].$$
> >
> > The same is true for the second order Taylor expansion around any energy function $f$:
> >
> > $$\Delta L_f[\delta f + c] = \Delta L_f[\delta f],$$
> > as can be easily verified from its expression (Equation 4).
> >
> > This is essentially the same as the well known property of the softmax function that one can add the same constant to all the input logits and it won't affect the output probabilities.
> > In other words, an energy function is only defined up to a global constant value.
> > Since $\delta f$ also only matters up to a constant, we are free to choose the $\delta f$ that has a certain expectation under some arbitrary distribution. Namely, we choose it so that it has 0 expectation under the current model to simplify the expressions.
> > Note that we could have made any other choice and it would lead to the same end-result. Only the expression for the leaf values would change by a constant value at the end (which doesn't matter when mapping to probabilities). Our choice simply makes the derivations simpler.
> >
> > > Why maximizing Equation 5 will improve the current energy function $f_t$ and how is $f_t$ defined and what is $t$?
> >
> > $f_t$ is defined in the text as the energy function at a hypothetical current boosting iteration.
> > You are right that we did not specify that $t$ would be the number of this boosting iteration. We will change the sentence above Equation 6 to read:
> >
> > - At each boosting iteration, $t$, we improve upon a current energy function $f_t$ by finding an optimal step $\delta f_t^*$ that maximizes $\Delta L_{f_t}$
> >
> > In other words, we take a local second order approximation to the log-likelihood at the current energy function, $f_t$, and maximize it to obtain a step $\delta f_t^*$. This is essentially finding a Newton step for the log-likelihood. Adding this step to the current energy function would therefore ideally improve the likelihood (as long as it is scaled appropriately).
> >
> > > How the size of $\mathcal{H}_t$ in Equation 6 impact their solution and how are they choosing a relatively small constrained set $\mathcal{H}_t$?
> >
> > At this point in the exposition we are not particularizing to any particular function space $\mathcal{H}_t$. This is done at the beginning of the following section where we explain our choice of $\mathcal{H}$ as a particular set of piecewise constant functions that partition the input space using binary trees. It is there that we explain how to carry out the optimization for this particular choice.
> > The size of this set is controlled by the hyperparameters of the weak learners, such as the number of leaves per tree, and these are selected by cross-validation.

---

> ### Author Response · Authors · 2024-11-22
> **Response (Part 2/2)**
>
> > **Figure 2**
>
> Again, this figure is only meant to provide a visual illustration of how NRGBoost's learned density can approximate a provided data distribution as $t$ increases and weak learners are added.
> The $\hat{p}(x)$ used in this instance is exactly what is depicted on the right side of the Figure. We are not drawing a sample from it since the point is not really to show generalization.
> In the limit $t \rightarrow \infty$, NRGBoost will converge to $\hat{p}(x)$, whether that is some empirical distribution or not.
> We use a simple 8-Gaussians 2D dataset where we can easily vizualize densities for this purpose.
> The radius of the circle on which the means of the Gaussians are placed is 8 times the standard deviation of the Gaussians (the global scaling factor doesn't really matter).
> While we don't really understand how knowing these specific details benefits the reader, we can add them to the caption.
>
> > It is confusing as the author put both discriminative and generative models in the same table
>
> This is motivated in the text but to summarize, the rationale is:
> - Good unconditional density models, $q(y, \mathbf{x})$, should also capture any given conditional distribution such as $q(y|\mathbf{x})$ well.
> - If a generative model performs better than another on a given discriminative task, this should provide some evidence that it is a better density model.
> - Discriminative models provide a strong baseline for the performance achievable since they fit $q(y|\mathbf{x})$  directly without needing to model $q(\mathbf{x})$ at all.
> - If an unconditional generative model matches this strong baseline this is particularly positive sign that it is a good model.
>
> >  As I am reading this comment, I am even more confused about how the author performed discrimination between class labels. They estimated an unnormalized density and nowhere in the paper, it is mentioned how they convert their class conditional density into posteriors and thereby, do the discrimination.
>
> Given an unconditional density model $q(y, \mathbf{x})$, all we do is compute the conditional by
>
> $$q(y \vert \mathbf{x}) = \frac{q(y, \mathbf{x})}{q(\mathbf{x})} = \frac{q(y, \mathbf{x})}{\sum_{y^\prime} q(y^\prime, \mathbf{x})}$$
>
> For a given sample $\mathbf{x}$, this requires evaluating $q(y, \mathbf{x})$ for every possible value of $y$ and then normalizing the obtained distribution over $y$.
> Note that this is essentially the same as Equation 2 for an EBM, as pointed out in our previous comment, where in this case $\mathbf{x}_u$ is $y$ and $\mathbf{x}_o$ is $\mathbf{x}$.
>
> > Related Work
>
> We did not move this section, we simply removed what was previously Section 4. As stated in our summary of changes:
>
> > Removed the section on our proposed improvements to Density Estimation Trees from the main paper and integrated it fully into **Appendix C**. We believe that this improves the flow of the paper by focusing on the main contributions. We are open to fully removing this contribution, as suggested by reviewer `NED1`, if the remaining reviewers agree.
>
> > I also do not see any pseudocode or flow diagram explaining the algorithm yet.
>
> We added a high level overview of our algorithm in Appendix A.3. We listed this under the Summary of changes comment above but forgot to mention it in our response.
>
> We hope this answers your questions.

---

> > ### Comment · Reviewer_a4Ct · 2024-11-22
> > **Thanks!**
> >
> > I thank the authors for providing such an elaborate rebuttal. I am raising my score and I am willing to raise more if my following concerns are satisfied:
> >
> > - MNIST: I did not say the authors are not treating the MNIST datasets as tabular dataset. I was trying to point out the fallacy of visualizing it as images. In the reference provided by the author, David S. Watson et al. exactly proved my point in the appendix (which is not even in their main text). They showed why they cannot outperform convolution based generative approaches on structured data. Whereas I guess the claim of the author here is different. They claim they can perform well on vision datasets although they do not consider the local correlation in the vision data. Note that the MNIST dataset is visualized in Figure1 which I guess they are trying to portray as a central claim. This is confusing from the perspective that they are proposing a generative model for tabular data. If they claim they do well even if they do not consider the local correlation, I will be satisfied if they replicate similar results on CIFAR10 or CIFAR100 (then I will raise my score even more). Otherwise, I would move the figure to appendix as an additional interesting result.
> >
> > - Equations: I would add the explanation to the equations to draft and refer to the appendix as necessary for details. Note that without referring to the appendix readers may not be prompted to examine the contents in the appendix. This also goes with the suggestion for pseudocode.
> >
> > - Figure 2: I would argue with the fact that the authors find it unnecessary to provide the details of the simulation performed. It will hinder the reproducibility of the paper contents for people who will try grok the paper in future.
> >
> > - Discrimination between class labels: I do not think their model is estimating $q(y,x)$, rather it is estimating $q(x | y)$. This makes me curious about the impact of priors in their model. Note that $q(y,x) = q(y) q(x|y)$. Are they considering $q(y)$ is equal for all the class labels? What happens if the dataset is unbalanced? In summary, estimation of $q(y|x)$ is not clear to me. I would not draw analogy between Equation 2 and the equation to estimate $q(y|x)$. Equation 2 has observed and unobserved variables whereas there is no such thing with $x$ and $y$.

---

> ### Author Response · Authors · 2024-11-24
> **Thank you!**
>
> Thank you for raising your score. We are pleased that our clarifications have improved your opinion of our work.
> We have uploaded a revised version of our paper, which we believe addresses your remaining concerns effectively.
>
> > They claim they can perform well on vision datasets although they do not consider the local correlation in the vision data. Note that the MNIST dataset is visualized in Figure1 which I guess they are trying to portray as a central claim.
>
> It was never our intention to claim that our model was suitable for image data or to imply that it can compete with specialized image models. In the caption we say:
> - Despite this being a simple image dataset, it can be challenging for tabular generative models due to the high dimensionality and complex structure of correlations between features. We find NRGBoost to be the only model that is able to generate passable samples.
>
> We see the MNIST task merely as a challenging tabular task that can be used to test tabular models (in the same way as the CTGAN paper which uses it as just another dataset to compare generative models).
> As a tabular model, NRGBoost needs to learn that features that correspond to nearby pixels are correlated or anti-correlated and has no helpful inductive bias for doing so, unlike image models.
> Despite this, we find that it accomplishes this task better than the other tabular models and, in particular, well enough that the samples it generates are coherent enough to resemble (passable) digits.
>
> That being said we agree with the reviewer that having this figure placed so early in the paper in page 2 can be confusing to a reader and give the wrong idea.
> **We have replaced this image with a diagram that provides an overview of NRGBoost's training process as you had previously suggested.**
>
> We still include the MNIST Figure in the experiments section, **after** we explain our motivation for including this dataset.
> We also improved the caption to emphasize that we do not expect NRGBoost to be competitive with image models:
> - Despite this being a simple dataset **that would pose no challenges to image models**, it is hard for tabular generative models due to the high dimensionality and complex structure of correlations between features.
>  We find NRGBoost to be the only **tabular** model that is able to generate passable samples.
>
> > Equations: I would add the explanation to the equations to draft and refer to the appendix as necessary for details. Note that without referring to the appendix readers may not be prompted to examine the contents in the appendix. This also goes with the suggestion for pseudocode.
>
> We added missing references to the Appendix and made small adjustments to the explanations.
>
> As for the algorithm pseudocode, we had already added a reference to it at the end of Section 3.2. We believe this to be the most appropriate place, since it is when the reader will have all the necessary context to fully understand the algorithm as it is described.
>
>
> > Figure 2: I would argue with the fact that the authors find it unnecessary to provide the details of the simulation performed. It will hinder the reproducibility of the paper contents for people who will try grok the paper in future.
>
> We added the details for reproducing Figure 2 to the Appendix on reproducibility details (and a reference to this appendix in the caption).
>
> > Discrimination between class labels: I do not think their model is estimating $q(y,x)$, rather it is estimating $q(x|y)$.
>
> We train all generative models to learn a distribution $q(\mathbf{z})$ over an input variable $\mathbf{z} = (y, \mathbf{x})$ that is the concatenated "target" and "input".
> The models do not treat $z_0$ (the target) is any special way different than the other $z_i$ (target and input are just labels that we give these variables and the model is in no way aware of them).
>
> As a result, the model is simply learning a joint distribution over $y$ and $\mathbf{x}$, not class conditional distributions and a separate class prior.
> In Section 5.1 we use these unconditional models to perform inference over $z_0$, conditioned on observed values of the remaining $z_i$ but we could just as well use them to infer the value of any other variable instead.
> This is why we say that these density models are flexible.
>
> We hope this answers your remaining questions.

---

> > ### Comment · Reviewer_a4Ct · 2024-11-24
> > **Thanks for the effort!**
> >
> > Thanks for your effort to address my concerns! I am raising my score further! However, I have a suggestion for the camera ready version. It will be helpful if the author can change the class priors in a simulation setup and plot the impact on their model. The simulation setup can be as simple as two Gaussian blobs representing two separate classes and gradually changing the class priors.

---

> ### Author Response · Authors · 2024-11-25
> **Thank you again!**
>
> Thank you for recognizing our efforts. We believe that our paper has improved as a result of this discussion.
>
> Regarding the class priors, if your concern is the impact of having a more balanced or unbalanced dataset, our classification datasets already have some variability in this respect:
> - MiniBooNE and Adult have prevalences of 28% and 24% for the minority classes respectively
> - MNIST is balanced across all 10 classes
> - Covertype is unbalanced across all 7 classes with the largest comprising ~49% of the dataset and the smallest only 0.5%
>
> For the regression datasets the $p(y)$ are also quite different from a uniform distribution over $y$.
> But we can try to set up a toy experiment to see if NRGBoost is any more affected by the base rates of each class than XGBoost in a binary setting.

---

> ### Author Response · Authors · 2024-11-30
> **Effect of class priors**
>
> Since Gaussian blobs would be too easy to classify we ran an additional simple experiment with the Adult dataset to check how NRGBoost's performance changes as we vary the class priors.
> We kept the total number of samples fixed (N=12516) but changed the prevalence of the two classes by subsampling from the original training data.
> We test the following values for the prevalence of the (original) minority class: 1%, 10%, 25% and 50% (i.e., balanced).
>
> For each value of the prevalence we train a XGBoost and a NRGBoost model (repeating 5 times with 5 different seeds for the subsampling).
> Below we report the AUC achieved by each model on the test set for each prevalence, averaged across seeds (with the respective standard errors):
>
> | prevalence | XGBoost       | NRGBoost      |
> |------------|---------------|---------------|
> | 0.01       | 0.890 (0.007) | 0.887 (0.003) |
> | 0.10       | 0.918 (0.002) | 0.911 (0.001) |
> | 0.25       | 0.923 (0.001) | 0.915 (0.001) |
> | 0.50       | 0.924 (0.000) | 0.918 (0.001) |
>
> Performance does get slightly worse as the datset becomes more unbalanced but the trend is very similar for both XGBoost and NRGBoost.
> This is also expected since for 1% prevalence there are only 125 examples of the minority class and the boundaries between the two classes become noisier.

---

### Official Review · Reviewer_pEfp · 2024-10-31

**Soundness:** 3
**Presentation:** 4
**Contribution:** 3
**Rating:** 6
**Confidence:** 3

**Summary:**

This manuscript presents a framework for using boosting to model unnormalised densities.
The proposed framework is used to build a model using decision trees as weak learners.
To accelerate sampling from the model (and thus training), a combination of Gibbs and rejection sampling is used.
Experiments on a down-sampled MNIST and UCI datasets demonstrate competitive performance to other tree-based generative models.

**Strengths:**

- (clarity) Very well-written paper that is easy to follow.
   I also appreciate the clear presentation and discussion of limitations of the proposed approach.
 - (originality) The proposed boosting framework seems to be novel and provides some refreshing insights.
 - (originality/quality) The overview over existing approaches to use decision trees for generative modelling is extensive and contextualises the work well.
 - (quality) The derivations are explained well and the appendix provides useful additional details.
 - (quality) The experiments seem to have been set up properly with error bars and competitive baselines.

**Weaknesses:**

- (significance) The abstract mentions a comparison with neural network models, but I could not find these in the main paper.
   Furthermore, the authors claim that tree-based generative models are to be preferred over DL-based models
   because discriminative DL methods do not provide competitive performance on tabular data.
   However, no DL methods have been included as baselines in the experiments to confirm this.
   As a result there is no evidence that DL methods would not be able to outperform tree-based generative models.
   PS: DL methods based on normalizing flows have explicit (tractable) densities.
 - (significance) It is not entirely clear how the proposed model can/should be used.
   The discussion mentions the overhead due to sampling, but this overhead is never quantified.
   This makes it hard to get a feeling for whether the (sometimes minor) increase in performance is "worth it".
   Also, it is not entirely clear why shallower trees are preferrable, as implied in the discussion.
 - (significance) The authors put a lot of emphasis on the tractability of computing densities.
   However, there are no clear experiments that illustrate the importance of this feature.
   The conclusion hints at applications enabled by density models,
   but this seems to indicate that there are no real use cases for this model.
   Especially since the results seem to indicate that there are other competitive methods out there.

**Questions:**

1. How does the performance compare to DL-based generative models?
2. Can the tradeoffs of NRGBoost vs e.g. DET/DEF be quantified somehow?
3. Is there a concrete use case where NRGBoost is the only/best model?

---

> ### Author Response · Authors · 2024-11-21
> **Author Response**
>
> Thank you for your valuable feedback and for acknowledging our contributions. We address your questions and concerns below.
>
> > **Comparison to DL-based models**
>
> Comparison between tree-based and DL-based *discriminative* methods on tabular data has been an ongoing debate with claims of superiority made on both sides.
> The goal of our paper is not to add to this discussion but to compare *generative* methods instead.
> We only provided a single discriminative baseline (XGBoost) as guidance for what performance should realistically be achievable on the task in Table 1 since discriminative models are specialized for this particular task.
> The DL methods that we refer to in the abstract are the generative baselines TVAE and TabDDPM which we compare to NRGBoost for sampling use cases.
>
> Research into DL based generative modeling has largely focused on models for which density estimation is untractable and, as a result, we are left with no generative DL methods to compare to in Table 1.
> We are not aware of any DL normalizing flow approaches but the fact that tabular data typicaly comprises a mix of continuous and discrete dimensions (be it ordered or not) would pose an obstacle to this type of approach. They would have to rely on dequantization of discrete numerical variables and encoding of categorical variables.
> The only DL generative model we are aware of that is capable of density estimation is the autoregressive approach of [1].
> Unfortunately, no code for this model is available making this comparison impractical.
>
> > **Relevance of tractable density estimation**
>
> This is fair criticism and we agree that we did not clearly motivate why one should care about density modeling approaches given that our choice of task in Section 5.1 particularly suits discriminative models.
> Our main goal was to show superiority of NRGBoost over other generative models but some of the advantages of generative density models over discriminative ones are:
> - They can do out of distribution detection.
> - There is prior work showing that generative classifiers can be more robust to distribution shifts or adversarial attacks [2][3][4].
> - Generative models allow for more flexible inference scenarios. As an example, they can be used to jointly predict the value of more than one variable.
> They can also handle prediction tasks with missing covariates, in a principled way, by marginalizing over unobserved variables.
>
> To better motivate the benefits of density models, we added a small case study of one such application of inference with missing data in **Appendix E.3**.
> There, we show that NRGBoost is able to outperform XGBoost with simple imputation strategies that are typically used by practitioners when one covariate is missing.
>
> > **Where is NRGBoost the best model?**
>
> Besides the previously mentioned density estimation scenarios,
> NRGBoost also outperforms the other baselines when looking at sample quality metrics and is generally close to TabDDPM in ML Efficiency.
>
> Note that the main downside of NRGBoost for sampling is simply that it is slow when independent samples are required since inexactness is already reflected in the reported performance metrics.
> We included a comparison of sampling times in this scenario in **Appendix E.6.2** in order quantify it. We find that NRGBoost is, in the median case, **10x slower for sampling than TabDDPM**, the only other method that can realistically compete in sample quality.
> We thus believe that NRGBoost is not only a very strong competitor for sampling use cases but also the most flexible generative model for tabular data overall.
>
> Another advantage of the energy-based approach that we don't explore in the paper is that it allows sampling from an arbitrary conditional distribution, requiring only that one clamps the observed values when Gibbs sampling. This contrasts with diffusion models which typically have to be trained to learn a specific conditional distribution.
> We have incorporated some of the above discussion into our Discussion section to hopefully make the motivation and relevance of NRGBoost clearer to the reader.
>
> > **Advantages of shallower trees**
>
> Inference from shallower and smaller models is faster which could be relevant in scenarios requiring low-latency.
>
>
> We hope our response adequately addresses your main concerns and welcome any further discussion.
>
>
> [[1]](https://arxiv.org/abs/2312.06089) Manbir S. Gulati and Paul F. Roysdon. TabMT: Generating Tabular data with Masked Transformers. In NeurIPS 2023
>
> [[2]](https://arxiv.org/abs/1802.06552) Yingzhen Li et al. Are Generative Classifiers More Robust to Adversarial Attacks? In ICML 2019
>
> [[3]](https://arxiv.org/pdf/2309.16779) Priyank Jaini et al. Intriguing Properties of Generative Classifiers. In ICLR 2024
>
> [[4]](https://openreview.net/pdf?id=02dpwytSRt) Alexander C. Li. Generative Classifiers Avoid Shortcut Solutions. In ICML 2024 Workshop SPIGM

---

> > ### Comment · Reviewer_pEfp · 2024-11-25
> > **Thank you for the replies**
> >
> > I must have missed the line that states that TVAE and TabDDPM are deep learning methods, sorry.
> > I also appreciate the experiment highlighting the possible relevance of the density estimation (even though it ended up in the appendix).
> >
> > I am afraid that I still do not quite see what the exact scenario/use-case for NRGBoost is.
> > The slower sampling and comparable quality seem to make it a hard sell.
> > Nevertheless, I do see the value of having this work published and correspondingly increase my score (from 5 to 6) to align with the other reviewers who also tend towards acceptance.

---

> > > ### Author Response · Authors · 2024-11-25
> > > **Thank you!**
> > >
> > > Thank you for raising your score and for recognizing the value in our work being published.
> > >
> > > In future work we hope to:
> > > - Explore alternatives to Gibbs sampling in order to improve what we agree to be one of the biggest downsides of NRGBoost.
> > > - Explore use cases that are enabled by density modeling or by flexible conditional sampling.
> > >
> > > We also believe that there is value in providing an alternative approach with a distinct set of strengths and weaknesses compared to current methods, as it may prove suitable for applications that we cannot yet anticipate.

---

### Official Review · Reviewer_cBTV · 2024-11-04

**Soundness:** 3
**Presentation:** 3
**Contribution:** 3
**Rating:** 6
**Confidence:** 3

**Summary:**

This paper explores generative extensions of tree-based models explicitly molding the data density for structured tabular data. Specifically, an energy-based generative boosting algorithm NRGBoost is proposed which is trained to maximize a local second order approximation to the likelihood at each boosting iteration. To save the major training budget caused by approximately estimating the event probability of input data with sampling, the authors further propose an amortized sampling approach by maintaining a fixed-size sample pool with rejection sampling at each round to reduce training time. Apart from designing NRGBoost, the authors also explore bagged ensembles of DET as a generative counterpart to Random Forest. Comprehensive experiments on tabular discriminative and generative (data synthetic efficiency & Discriminator Measure) tasks show NRGBoost can be comparable to the prevailing discriminative GBDTs on prediction tasks as well as has competitive generation quality compared to recent tabular generative neural models, all achieved in one gradient boosting algorithm.

**Strengths:**

I am not expert in the fields of generative models or tree-based models, therefore I think I am not qualified to comment on the paper novelty, while there do exist certain aspects making me impressive.

**Solid formulation, proof & demonstration**: Based on the theory foundation of GBDT, the extension to its generative version NRGBoost is natural and its formulation derivation is clear and convincing. The proposed amortized sampling approach is reasonably designed utilizing the properties of the boosting algorithm. The experiments on tabular discriminative & generative tasks are well conducted with repeated trails, statistical tests and visualization analysis. Besides, complete theoretical and technical Appendix is given, including the computational time comparison for training each baseline.

**Wide application scenarios**: As a generative boosting algorithm, NRGBoost is able to be applied to both discriminative and generative tasks, which distinguishes it from other neural-based generative baselines.

**Reference-worthy exploration on tree-based models**: This paper explores the generative extensions in both Boosting- and Bagging-based tree models, giving the reference for the further research in the community.

**Weaknesses:**

Several weaknesses from application perspective.

(1) From Table 1, in discriminative tasks the performance gap between NRGBoost and traditional discriminative GBDTs (e.g., XGBoost) seems to be relatively more significant as the data scale increases, there is still room for further improvement before application in large-scale discriminative classification tasks.

(2) From the computational effort analysis during training, in Fig. 4, it seems NRGBoost is not computation-economical in the large datasets compared to other neural-network-based generative models, the data scalability of NRGBoost is not sufficiently discussed.

(3) From evaluated dataset information in Table 5, there lacks large regression datasets, which help us to further realize NRGBoost on large-scale regression tasks.

**Questions:**

I would like to increase my score according to the comments from other reviewers and the response.

(1) What about the inference time comparison between NRGBoost and other baselines?

(2) Why there is often a performance gap between discriminative GBDTs (i.e., XGBoost here) and NRGBoost? Is it possible to outperform these discriminative baselines from generative modeling paradigm?

(3) How about the discriminative performance of NRGBoost on large regression datasets (e.g., Microsoft, Yahoo dataset in [1]), large-feature-amount one (e.g., Epsilon in [1]) and large-class-number one (e.g., ALOI in [1])?

**Reference**

[1] Revisiting Deep Learning Models for Tabular Data, NeruIPS 2021.

---

> ### Author Response · Authors · 2024-11-21
> **Author Response (Part 1/2)**
>
> Thank you for your thoughtful feedback and for recognizing the relevance of our work. We address your questions below.
>
>
> > **Inference Times:**
>
> Evaluating the energy or density of a point for a tree-based generative model requires evaluating on which leaf falls each example and is therefore similar to evaluating any other discriminative tree-based model like XGBoost.
> Evaluating the conditional density over a single target variable can similarly be efficiently evaluated with a single pass over each tree in the ensemble, branching on splits where the splitting variable is the target variable.
> In practice, for NRGBoost, this is quite fast, and it predicts the entire test set for the Covertype dataset (with 16384 leaves per tree) in 3.8 seconds (single-threaded), and for MNIST in 0.4 seconds.
>
> Note that this inference process is similar for NRGBoost, DEF and ARF, so any differences can mostly be attributed to size of the models. DEF models are larger and therefore slower and ARF models fit non-uniform distributions at the leaves that need further evaluation.
>
> > **Performance Gap between NRGBoost and Discriminative Models**
>
> This is an interesting question.
> In Section 5.1, we are training unconditional density estimators to model the full $p(\mathbf{x}, y)$, which requires learning the relationships between all variables.
> Then we are using these flexible models to infer the value of a single specific input variable $y$.
> In contrast, discriminative methods are specialized for this particular task and cannot perform inference over any other variable. They only need to model the relationship, $p(y\vert \mathbf{x})$, between this target variable and the others, without learning $p(\mathbf{x})$ at all.
>
> Note, for example that, for a tree based generative model, any branch of a tree that doesn't contain at least one split on the target variable $y$ does not add any discriminative power about $y$.
> In contrast, every split added to a discriminative tree-based model is aimed at improving $p(y\vert \mathbf{x})$ since the splits on $y$ are already implicit.
> As a result, we do not believe it is reasonable to expect that unconditional generative models would consistently outperform discriminative ones in this setting.
> We merely wanted to show that NRGBoost can learn unconditional density models that are good enough to even be comparable to SoTA specialized discriminative models in some scenarios.
>
> We consider having no performance gap as a success already since, besides sampling, NRGBoost can handle more complex inference tasks such as joint inference over more than one variable, or prediction with missing covariates by marginalizing over unobserved variables. We added one such example in **Appendix E.3** showing that by marginalizing over a missing variable NRGBoost can outperform XGBoost with simple imputation approaches.
>
> If one really wanted better performance in inference over a particular variable $y$, one can learn class conditional generative models $p(\mathbf{x} \vert y)$ or, similarly, force a full early initial split on $y$, on every tree.
> This would be a fairer comparison, and one where prior work ([1][2]) has shown that generative classifiers can be more robust to distribution shifts or adversarial attacks.
>
> [[1]](https://arxiv.org/abs/1802.06552) Yingzhen Li et al. Are Generative Classifiers More Robust to Adversarial Attacks? In ICML 2019
>
> [[2]](https://openreview.net/pdf?id=02dpwytSRt) Alexander C. Li. Generative Classifiers Avoid Shortcut Solutions. In ICML 2024 Workshop SPIGM

---

> ### Author Response · Authors · 2024-11-21
> **Author Response (Part 2/2)**
>
> > **Training Scalibility**
>
> NRGBoost, similar to other tree-based methods, scales linearly in both the number of samples and the number of features.
> Neural network methods would exhibit the same linear scaling in number of samples if they were trained for a fixed number of epochs but the scaling in the number of features depends on the model architecture.
> Note however that what we report in Fig. 4 are the training times for the models selected by hyperparameter tuning.
> For the larger datasets, larger NRGBoost models with higher number of leaves per tree performed comparatively better which partially helps explain the longer training times.
> Notably, the performance gap between NRGBoost and the neural network models was also largest in the MNIST and Covertype datasets.
>
>
> > **Performance Gap on Challenging tasks**
>
> We have included in **Appendix E.4** some extra results on the requested datasets:
> - **ALOI:** Unfortunately our implementation of NRGBoost is currently limited to discrete features with a cardinality of up to 255. While this restriction could be easily removed later, we decided, as a compromise, to include only the first 250 classes out of the original 1000 in this dataset. We believe this still effectively tests the scaling to larger number of classes, given that it is 25 times the number of classes in MNIST. We find that in this dataset, NRGBoost actually outperforms XGBoost (0.962 vs 0.943 accuracy).
> - **Microsoft:** This dataset is quite challenging since it is already 1 order of magnitude larger than our largest datasets (in terms of the product `n_samples * n_features`). As a result we had to resort to manual tuning as our hyperparameter tuning protocol was unpractical. The best model we were able to find in this limited time still had a reasonable gap in performance, achieving a test RMSE of 0.787 compared to 0.744 for XGBoost and took 2 hours and 36 minutes to train.
> - **Epsilon:** With this dataset being nearly 2 orders of magnitude larger than Covertype and MNIST, we did not find it practical to use it for experiments. We believe that the setting of high dimentionality of the input to be particularly challenging for tabular generative models in general.
>
> As we explained in our previous comment, we don't expect unconditional density models to consistently outperform or even match discriminative ones.
> We also aknowledge that this gap can become more pronounced the larger the dataset becomes, as the results on the Microsoft dataset would seem to indicate.
>
> We would like to note, however, that we are unware of prior works on tabular generative models using datasets larger than Covertype or MNIST.
> Furthermore, there are certainly opportunities to optimize our implementation further to better tackle these large datasets, particularly our tree-fitting code which is single-threaded at the moment.
> Note that here we are comparing to a industry standard boosting framework with many engineer-hours put into optimizing it.
>
>
>
>
> We hope to have adequately addressed your main concerns and we welcome any further discussion.

---

> > ### Comment · Reviewer_cBTV · 2024-12-02
> > **Thank you for the clear response**
> >
> > Thanks for the point-by-point response which sufficiently addressed my concern, I hold my positive evaluation.

---

> > > ### Author Response · Authors · 2024-12-02
> > > **Thank you**
> > >
> > > Thank you again for your feedback. We are glad we were able to address your concerns.

---

### Comment · Area_Chair_BRZq · 2024-11-13
**authors - reviewers discussion open until November 26 at 11:59pm AoE**

Dear authors & reviewers,

The reviews for the paper should be now visible to both authors and reviewers. The discussion is open until November 26 at 11:59pm AoE.

Your AC

---

### Author Response · Authors · 2024-11-21
**Summary of changes**

We thank the reviewers for their time and constructive feedback.

We are pleased that, with the exception of one particularly critical reviewer, the reviewers generally found our work to be clearly presented and insightful. Additionally, reviewers recognized the relevance and significance of our contributions.

The main concerns raised by the reviewers related to motivation for density models and requests for additional experiments and results.
We hope that the revised version of our manuscript adequately addresses these concerns.
The main changes we made are:
- Included a new experiment in **Appendix E.3** that highlights one of the benefits of the flexible inference capabilities of density models in handling missing covariates. We hope this compelling example underscores the value of generative density models.
- Improved the *Background* section and the *Discussion* to provide additional motivation for density models, in order to address feedback from reviewer `pEfp`.
- Added NGBoost as a second discriminative baseline for regression tasks, which, like NRGBoost is capable of uncertainty quantification, per reviewer `NED1`'s request.
- Added an analysis on uncertainty estimation in **Appendix E.2**, showing that NRGBoost outperforms this discriminative baseline in all of our regression tasks.
- Removed the section on our proposed improvements to Density Estimation Trees from the main paper and integrated it fully into **Appendix C**. We believe that this improves the flow of the paper by focusing on the main contributions.
We are open to fully removing this contribution, as suggested by reviewer `NED1`, if the remaining reviewers agree.
- Added a discussion on relevent recent and concurrent work to the *Related Work* section as requested by reviewer `NED1`.
- Added a comparison in **Appendix E.4** between NRGBoost and XGBoost on two challenging tasks requested by reviewer `cBTV`.
- Added a high-level algorithmic description of NRGBoost in **Appendix A.3** to address feedback from reviewer `a4Ct`.
- Improved the caption of Figure 2 to address feedback from reviewer `a4Ct`.
- Implemented small changes to the text suggested by reviewer `NED1` and removed colloquialisms and abbreviations pointed out by reviewer `a4Ct`.

The main additions to the manuscipt are highlighted in blue for clarity.
We hope these revisions address the reviewers’ concerns and significantly improve our work. Thank you again for your thoughtful feedback.

---

> ### Author Response · Authors · 2024-11-24
> **Summary of changes (v2)**
>
> Dear reviewers,
>
> We have uploaded a new version of our paper in order to address some additional feedback from reviewer `a4Ct`.
>
> The following are the additional changes made over the previous version:
> - Replaced Figure 1 (MNIST) with a diagram depicting the training loop for NRGBoost.
> - Moved the MNIST Figure to the experiments section and adjusted its caption.
> - Added missing references to the Appendix throughout the text.
> - Adjusted a few sentences in **Section 3**.
> - Added additional details of the toy data used in Figure 2 to **Appendix D.1** and updated the caption.
>
> Again, the main additions and changes are highlighted in blue.
>
> Thank you again for your time.

---

### Meta-Review · Area_Chair_BRZq · 2024-12-19

**Metareview:**

The paper extends Gradient Boosted Decision Trees and Random Forests to generative modeling tasks. This paper went through significant changes during the review period, in a persistent effort from the authors to address comments from the reviewers, who eventually increased their original scores. My overall impression from the paper itself and the reviews and discussion is also positive. Modelling density is not a natural task for forests and this paper adds value to the small literature on this topic. It also may attract interest to the use of forests in generative modelling. I do concur with a reviewer that it is not clear what the exact use case is for NRGBoost, though.

**Additional Comments On Reviewer Discussion:**

N/A

---

### Decision · Program_Chairs · 2025-01-22

Accept (Poster)